# SubDiff: Subgraph Latent Diffusion Model

## Abstract

Diffusion models have achieved impressive performances on generative tasks in various domains. While numerous approaches are striving to generate feature-rich graphs to advance foundational science research, there are still challenges hindering generating high-quality graphs. First, the discrete geometric property of graphs gains difficulty in capturing complex node-level dependencies for diffusion model. Second, there is still a gap to simultaneously unify unconditional and conditional generation. In this paper, we propose a subgraph latent diffusion model to jointly address above challenges by inheriting the nice property of subgraph. Subgraphs can adapt diffusion process to discrete geometric data by simplifying the complex dependencies between nodes. Besides, subgraph latent embedding with explicit supervision can bridge the gap between unconditional and conditional generation. To this end, we propose a subgraph latent diffusion model (SubDiff) by taking subgraphs as minimum units. Specifically, a novel Subgraph Equivariant Graph Neural Network is proposed to achieve graph equivariance. Then a Head Alterable Sampling strategy (HAS) is devised to allow different sampling routes along diffusion processes, unifying the conditional and unconditional generative learning. Theoretical analysis demonstrate that our training objective is equivalent to optimizing the variational lower bound of log-likelihood. Extensive experiments show SubDiff achieving better performance in both generative schemes.

## 1 Introduction

Diffusion models (DMs), as a novel generative paradigm, have achieved numerous impressive results in the image and text generation (Ho et al., 2020; Song et al., 2020; Dhariwal & Nichol, 2021; Li et al., 2022). Inspired by non-equilibrium thermodynamics theory, DMs model the learning process as Markov chains trained with variational inference (Sohl-Dickstein et al., 2015). They consist of two stages (chains), i.e., forward diffusion and reverse denoising. The forward process gradually adds noise to original data, while the backward process reverses the noise by a learnable denoising neural network. The generative process is to feed random noise and recover to a well-trained denoising neural network. Prior practices of DMs are mainly concentrated in the domains of image and text. Recently, graph diffusion models have been extensively studied to advance foundational science research (Wu et al., 2022; Xu et al., 2023). However, there are still some challenges that have not been well solved, i.e., the difficulty in modeling discreteness of geometric data and the gap between unconditional and conditional generative learning.

The discreteness of geometric data introduces tremendous challenges to diffusion models on graph domain (Fan et al., 2023). Actually, the key problem lies in that the complex intrinsic dependencies among nodes are non-trivial and not explicit to be captured by generative frameworks. Previous studies with node-level diffusion tend to destroy the rich semantic dependencies of the original graph (Xu et al., 2022; Hoogeboom et al., 2022; Xu et al., 2023). Empirical evidence has verified this view in Appendix A.2. We observe most invalid samples generated by node-level generative models trap into the insufficient understanding of node-wise connections. We argue that the reason is that graph generative models generate not only the features of each node but also the complex semantic association between nodes. Therefore, the reason can lie in that the diffusion of each node or edge cannot be modeled as an independent event.

Besides, there still exists huge gap between unconditional (Un-G) and conditional (Con-G) generative models (Ho & Salimans, 2022; Liu et al., 2023; Fan et al., 2023). Previous works learn Un-G and Con-G respectively by twice-trained paradigm. The reason for this training pattern is that they have

their own training expectations. Un-G focuses on the diversity of generated samples, while Con-G is expected to generate samples with desired properties. Actually, these two generative schemes should not be contradictive. Con-G is essentially the resampling of desired property from Un-G sampling space, and the space of Un-G is the combination of all conditionally generated sampling space. However, pervious independent training paradigm fails to bridge the mutual relationships between Un-G and Con-G.

**Present work.** We propose a subgraph-level latent diffusion model (SubDiff) to address these challenges. SubDiff takes subgraph latent embedding as the minimum unit in diffusing and sampling to overcome the complex intrinsic dependencies between nodes, and further proposes a Head Alterable Sampling strategy (HAS) to unify Un-G and Con-G models.

Subgraph has natural advantages to accommodate discrete non-Euclidean geometries (Zhao et al., 2021; Yang et al., 2023; Zhang et al., 2021). On the one hand, subgraph-level modeling can simplify most connections between nodes, which can alleviate the challenges posed by complex dependencies in graphs (Yu & Gao, 2022). On the other hand, taking subgraph as a minimum unit of graph is a common inductive bias in various science domains, such as functional groups in molecules, which benefits exploring more interpretable generative models. However, bringing subgraphs into diffusion learning scheme raises two critical questions, i.e., *how to extract subgraph* and *how to guarantee subgraph-level equivariance*. We first summarize that the veracity of subgraph and the sufficiency of subgraph set are the key to extracting subgraphs. Thus, we employ a frequency-based subgraph extractor following MiCaM (Geng et al., 2023), and we propose a subgraph-level equivariant architecture (SE-GNN).

We attempt to unify Un-G and Con-G by predefining appropriate sampling sapce. Based on this, we design a head-alternable sampling strategy in our generative model, to achieve such unification. Specifically, labels can be exploited as an explicit supervision during the training process of Autoencoder (AE). Different from traditional latent diffusion models, supervised information is conducive to obtaining interpretable subgraph latent embedding with meaningful property. This intuition empowers a great potential to integrate condition into unconditional models. To this end, we first investigate how to map the latent embedding space with the supervision, and then devise a Head Alterable Sampling strategy for integrating unconditional and conditional generation into one unified model.

Our study achieves following observations and contributions. First, we propose a paradigm of subgraph-level latent diffusion (SubDiff) to counteract the discreteness challenge in graphs generation. Then, a frequency-based subgraph extractor Geng et al. (2023) is utilized and a novel subgraph equivariant framework is propsoed to encode subgraph latent embedding. Secondly, we unify the unconditional and conditional generation models from the perspective of sampling space. HAS is proposed and various empirical results verify its excellent performance. Besides, theoretical analysis shows that our training objective is equivalent to optimizing the variational lower bound of the log-likelihood. Third, we conduct detailed evaluations on multiple benchmarks. All empirical results demonstrate that SubDiff is with superior capacity to accommodate graphs, and achieves a unified generative learning paradigm.

## 2 RELATED WORK

**Generative Models on Graph.** Driven by recent advances in Deep Neural Networks (DNNs) techniques, deep generative models, such as variational autoencoder (VAE) Simonovsky & Komodakis (2018), Generative Adversarial Networks (GAN) De Cao & Kipf (2018) , and normalizing flows Luo et al. (2021), have largely improved the generation performance for graph-structured data. To further improve the generative modeling capacity, diffusion models are proposed and widely used in graph generation tasks (Peng et al., 2023; Wu et al., 2022; Austin et al., 2021). Among them, the stable (latent) diffusion model is verified to have superior performance (Rombach et al., 2022; Xu et al., 2023). Actually, all diffusion models are node-wise operations. The only difference among numerous practices is whether the diffusion process based on specific graph structure or node-level embedding. There are few methods considering the effect of complex dependencies between nodes on diffusion process (Fan et al., 2023). Besides, there still exists huge barrier between unconditional and conditional generation, where a joint unconditional-conditional learning task follows a multi-time training paradigm. Along the line of diffusion models, we propose to alleviate the influence of

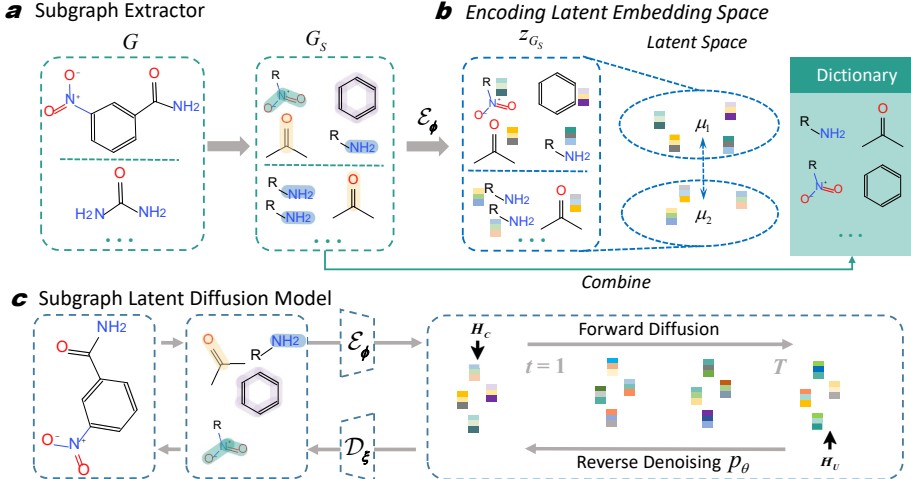

Figure 1: The architecture of SubDiff. **a:** Subgraph extractor aims to find high-frequency subgraphs in each graph and combine into a subgraph dictionary. **b:** The latent embedding space of two graphs with opposite solubility characteristics (*LogP*). **c:** Our subgraph latent diffusion model achieves Head Alterable Sampling, i.e., $H_C$ and $H_U$ denotes conditional and unconditional generation head respectively.

complex node-level dependencies, and bring unconditional and conditional generative patterns into a unified training paradigm.

**Subgraph Learning.** Subgraph learning aims to extract a bag of subgraphs to enable models to achieve more powerful representation ability (Yang et al., 2023; Frasca et al., 2022). This idea is also used to advance fundamental science research (Geng et al., 2023). The key consensus of these researches is that subgraph-level equivariance is vital in learning subgraph embedding (Bevilacqua et al., 2021). In this work, based on this intuition, we present a subgraph-level equivariance architecture to achieve subgraph diffusion model.

## 3 PRELIMINARY

**Notations.** In this paper, we consider undirected graphs $G = (\mathcal{X}, \mathcal{E})$, where $\mathcal{X}$ denotes the feature all nodes and $\mathcal{E}$ represents the connection information of nodes. $G_S$ is the set of subgraphs extracted by $G$. $z_G$ and $z_{G_S}$ represent graph latent embedding and subgraph-level representation set respectively.

### 3.1 PROBLEM DEFINITION

The challenges to graph latent diffusion models are two-fold. First, the discreteness of graph data increase the difficulty in comprehending the complex node-level dependencies. Second, the unification of unconditional and conditional generative schemes is the second challenge. Therefore, we consider the following two targets in this paper.

**(I) Achieve subgraph-level generation process.** Subgraph-level study can directly model the inherent substructure thus detouring the complex dependencies between nodes. Given a graph $G$, our goal is to achieve subgraph-level diffusion and denoising, which raises an even challenge of extracting subgraphs.

**(II) Unify unconditional and conditional generative patterns.** Different from previous twice-trained generative models, our goal is to unify the two generative paradigms into one model. This calls for finding out the commonality and unified learning scheme via embedding space.

## 3.2 DIFFUSION MODELS

Diffusion models (DMs) learn data distributions by modeling the reverse of a diffusion process, which constructs two Markov chains. (Xu et al., 2023; Ho et al., 2020) They respectively diffuse the data with predefined noise and reconstruct the desired samples from the noise.

In the forward diffusion chain, DMs gradually add Gaussian noise to data $x^{(t)}$ from raw data distribution $x_0 \sim q(x_0)$ for $t = 1, ..., T$, where $x_0$ denotes $G$ in graph diffusion model.

$$q(x^{(t)}|x^{(t-1)}) = \mathcal{N}(x^{(t)}; \sqrt{1 - \beta_t}x^{(t-1)}, \beta_t I), \quad q(x^{(1:T)}|x^{(0)}) = \prod_{t=1}^{T} q(x^{(t)}|x^{(t-1)}) \quad (1)$$

where $\beta_t \in [0, 1]$ represents the variance of the Gaussian noise added at time step $t$.

In the reverse denoising chain, DMs reconstruct original data from $x^{(T)}$ using $p_\theta$ parameterized by $\theta$.

$$p_\theta(x^{(t-1)}|x^{(t)}) = \mathcal{N}(x^{(t-1)}; \mu_\theta(x^{(t)}, t), \rho_t^2 I), \quad p_\theta(x^{(0:T)}) = p(x^{(T)}) \prod_{t=1}^{T} p_\theta(x^{(t-1)}|x^{(t)}) \quad (2)$$

where the $\mu_\theta$ is neural network, and the variance $\rho_t$ is also predefined. Since $q(x^{(1:T)}|x^{(0)})$ can be viewed as a fixed posterior, the learning process of DMs is to train $p_\theta$.

The variational lower bound of the likelihood of the data is given by,

$$\log p(x^{(0)}) \geq \underbrace{\log p_\theta(x^{(0)}|x^{(1)})}_{\mathcal{L}_{reconstruct}} \underbrace{-D_{KL}(q(x^{(T)}|x^{(0)})||p(x^{(T)}))}_{\mathcal{L}_{prior}}$$
$$\underbrace{-\sum_{t=2}^{T} D_{KL}(q(x^{(t-1)}|x^{(t)}, x^{(0)})||p_\theta(x^{(t-1)}|x^{(t)}))}_{\mathcal{L}_{denoise}} \quad (3)$$

Thus, $p_\theta$ is trained to maximize the variational lower bound of the likelihood of over all samples,

$$\mathcal{L}_{vlb} = \mathcal{L}_{reconstruct} + \mathcal{L}_{prior} + \mathcal{L}_{denoise} \quad (4)$$

## 3.3 SUBGRAPH LEARNING

Subgraph learning aims to extract a bag of subgraphs $G_S = \{G_S^1, ..., G_S^k\}$ to alleviate the deficiency caused by node-wise graph representation methods (Frasca et al., 2022; Miao et al., 2022; Yang et al., 2023). Actually, this practice is proved to be more expressive and interpretable both theoretically and empirically. Specifically, predefined structure and learnable discovery are two lines of research for subgraph extraction. The former achieves outstanding performance in the theoretical research of model expressiveness and some domain-special subgraph learning methods. The latter focuses more on analyzing the causal relationship between subgraphs and labels.

It is known that equivariance plays a significant role in graph representation (Bevilacqua et al., 2021). Similarly, subgraphs, which extracted from a whole graph, are faced the same equivariance issue. To this end, if $f(\tau \cdot G_S) = \tau \cdot f(G_S)$ always holds on for any permutation $\tau$ acting on subgraphs, then $f$ is equivariant. In molecular generation task, we focus on the Euclidean group E(3) generated by translations and rotations in 3D space (Xu et al., 2023).

## 4 SUBDIFF: SUBGRAPH LATENT DIFFUSION MODEL

In this section, we introduce the subgraph latent diffusion model (SubDiff), which consists of two major components, i.e., Subgraph-level Equivariant Graph Neural Network (SE-GNN) for encoding and Head Alterable Sampling (HAS) for unifying unconditional and conditional generative schemes.

## 4.1 Subgraph-level Encoding in Diffusion Process

The discrete nature of graphs, different from image data with continuous space, blocks the exploration of diffusion models in geometry domain. Subgraphs have natural advantages for discrete non-Euclidean geometries (Miao et al., 2022; Zhao et al., 2021). We propose subgraph-level latent diffusion model, which takes the subgraphs latent embedding as the minimum unit for diffusing and denoising. The use of subgraphs gives rise to a series of new questions, i.e., *how to extract subgraph* and *how to guarantee subgraph-level equivariance*. Next, we will gradually answer them.

During the generation process, we should determine the type set (dictionary) of all elements, such as atoms type in node-level molecules generation. This requires us to extract interpretable and sufficient subgraphs. Thus, we point out that the veracity of each subgraph and the sufficiency of subgraph dictionary should be primarily guaranteed in the stage of subgraph discovery. The veracity refers to the extracted subgraphs should be domain-meaningful, while the sufficiency means that we should extract all domain-related subgraphs to support subgraph dictionary. In the case of molecular generation, it is very significant to extract meaningful functional group (subgraph) structure and build a sufficient subgraph dictionary. In our work, we follow MiCaM (Geng et al., 2023), a frequency-based subgraph extractor, to guarantee veracity and sufficiency. This method identifies the highest frequency subgraph pairs, and merge them into an entire fragment (subgraph). Compared with the learnable subgraph extraction methods, frequency-based strategies enjoys the nice capacity to extract accurate and sufficient subgraphs.

Although subgraph can effectively detour most complex connections between nodes, subgraph-level equivariance still needs to be considered. Given the subgraph set $G_S = \{G_S^1, G_S^2, ... G_S^k\}$ extracted by $G$, Equation 5 should always hold on for any subgraph-level permutation operation $\tau$.

$$\tau \cdot \mathcal{E}_\phi(G_S) = \mathcal{E}_\phi(\tau \cdot G_S), \quad \tau \cdot \mathcal{D}_\xi(\mathbf{z}_{G_S}) = \mathcal{D}_\xi(\tau \cdot \mathbf{z}_{G_S}) \tag{5}$$

**Subgraph Equivariant Graph Neural Network.** We propose the equivariant subgraph learning architecture SE-GNN. It consists of the siamese component ($L^1$) and the information sharing component ($L^2$). The encoder $L^1$ independently acts on each subgraph, while $L^2$ forces to capture sharing information across subgraphs via taking reconstructed graph with subgraph as input. In practice, SE-GNN can also support the equivariance properties of E(3) symmetries, i.e., 3D translation and rotation. More details are shown in Section 5.3.

$$(L(\mathcal{X}, \mathcal{E}))_i = L^1(\mathcal{X}_i, \mathcal{E}_i) + L^2(\mathcal{X}^S, \mathcal{E}^S) \tag{6}$$

$(L(\mathcal{X}, \mathcal{E}))_i$ represents the output of the layer on the $i$-th subgraph. $(\mathcal{X}_i, \mathcal{E}_i)$ denotes the $i$-th subgraph element, and $(\mathcal{X}^S, \mathcal{E}^S)$ indicates reconstructed graph with the unit of subgraph.

## 4.2 Unifying Unconditional and Conditional Generative Model

The diversity and specificity of sampling space are the focuses of unconditional and conditional generative schemes respectively, which leads to their different learning preferences. In other words, unconditional generative models aim to obtain broader sampling space $\Omega^U$ via $p_\theta : \Omega^N \to \Omega^U$, while conditional sampling is more focused on acquiring more specific sampling space $\Omega^{C_i}$ by $p_\psi : (\Omega^N, y_i) \to \Omega^{C_i}$. We define $\Omega^{C_i}$ to represent the sampling space with $y_i$ as the condition. $\Omega^N$ represents the standard Gaussian space, and $\Omega^U$ denotes the unconditional sampling space.

However, we argue that these two generative processes should not be contradictive. The conditional generative space can be viewed as the conditional distribution $p(\Omega^{C_i}|\Omega^U, y_i)$, while unconditional sampling space is the union of all properties spaces. Previous twice-trained generation models are difficult to build a bridge between unconditional and conditional schemes, due to their independent training parameters and different learning preferences. Therefore, we propose an insightful proposition from the perspective of sampling space to unify these two generative schemes.

**Proposition 1.** *Let $\Omega^U$ and $\Omega^{C_i}$ be unconditional and $y_i$-conditioned sampling spaces respectively, where $i = 1, .., m$. If there exists $\Omega^{C_i} \subset \Omega^U$ and $\bigcup_{i \in [m]} \Omega^{C_i} = \Omega^U$, then unconditional and conditional generative models can be unified.*

This proposition delivers that the efforts from the perspective of sampling space have the potential to build a bridge between unconditional and conditional generative schemes. Now we provide a practical implementation with HAS to achieve their unity.

In the training stage, we propose a unified diffusion model with supervised information. Specifically, the label information is used as the prior constraint to obtain domain-specific subgraph latent representation in the training process of AE. Different from variational autoencoder (VAE) using standard Gaussian to constrain latent representation (Simonovsky & Komodakis, 2018), we personalize the prior distribution $\mathcal{N}(\mu_i, I)$ for each property $y_i$ to constrain graph latent embedding. In other words, we expect each sample with label $y_i$ satisfying

$$\mathbf{z}_G = \text{pooling}(\mathbf{z}_{G_S}) \sim \mathcal{N}(\mu_i, I) \tag{7}$$

where pooling is MEAN operation. This enables the diffusion process to be guided by a supervised latent representation, which is also designed as the beginning of conditional generative process in our work. Thus, the practical training objective of AE is

$$\mathcal{L}_{AE} = \mathcal{L}_{recon} + \mathcal{L}_{reg} = -\mathbb{E}_{p_\phi(\mathbf{z}_{G_S}^{(0)}|G)}[\log p_\xi(G|\mathbf{z}_{G_S}^{(0)})] + D_{KL}(p_\phi(\mathbf{z}_G|G)||p(\mathbf{z}_G)) \tag{8}$$

We extract subgraphs and obtain subgraph-level latent representation. Next, we will discuss the diffusion and denoising process of SubDiff. Analogous to traditional latent diffusion models, we can train the model $p_\theta$ by

$$\mathcal{L}_{LDM} = \mathbb{E}_{\varepsilon \sim \mathcal{N}(0,I),t}[w(t)||\varepsilon - \varepsilon_\theta(x^{(t)}, t)||^2] \tag{9}$$

where $w(t)$ is set to 1 for each $t$.

The combined objective of SubDiff could be $\mathcal{L} := \mathcal{L}_{LDM} + \mathcal{L}_{AE}$, and we provide a theoretical analysis to prove that $\mathcal{L}$ is the variational lower bound of the log-likelihood.

**Theorem 1.** *Let $\mathcal{L} := \mathcal{L}_{LDM} + \mathcal{L}_{AE}$. $\mathcal{L}$ is a variational lower bound to the log-likelihood , i.e., there is hold on for any $G$,*

$$\mathbb{E}_{p_{data}}[\log p_{\phi,\xi,\theta}(G)] \geq -\mathcal{L} \tag{10}$$

*Proof.* We denote $\mathbf{z}_{G_S}^{(0)}$ as the original latent representation of subgraph set $\mathbf{z}_{G_S}$. Then we have

$$
\begin{aligned}
\mathbb{E}_{p_{data}}[\log p_{\phi,\xi,\theta}(G)] &= \mathbb{E}_{p_\phi(\mathbf{z}_{G_S}|G)}[\log \frac{p_\xi(G|\mathbf{z}_{G_S}^{(0)})p(\mathbf{z}_{G_S}^{(0)})}{p_\phi(\mathbf{z}_{G_S}^{(0)}|G)}] \\
&\geq \mathbb{E}_{p_\phi(\mathbf{z}_{G_S}^{(0)}|G),q(\mathbf{z}_S^{(1:T)}|\mathbf{z}_S^{(0)})}[\log \frac{p_\xi(G|\mathbf{z}_{G_S}^{(0)})p(\mathbf{z}_{G_S}^{(0:T)})}{p_\phi(\mathbf{z}_{G_S}^{(0)}|G)q(\mathbf{z}_S^{(1:T)}|\mathbf{z}_S^{(0)})}] \\
&= \mathbb{E}_{p_\phi(\mathbf{z}_{G_S}^{(0)}|G)}[\log p_\xi(G|\mathbf{z}_{G_S}^{(0)})] - D_{KL}(q(\mathbf{z}_{G_S}^{(T)}|\mathbf{z}_{G_S}^{(0)})||p(\mathbf{z}_{G_S}^{(T)})) \\
&\quad + \mathbb{E}_{p_\phi}\log \frac{q(\mathbf{z}_{G_S}^{(T)}|\mathbf{z}_{G_S}^{(0)})}{p_\phi(\mathbf{z}_{G_S}^{(0)}|G)} - \sum_{t=1}^{T} D_{KL}(q(\mathbf{z}_{G_S}^{(t)}|\mathbf{z}_{G_S}^{(t-1)}, \mathbf{z}_S^{(0)})||p_\theta(\mathbf{z}_{G_S}^{(t-1)}|\mathbf{z}_{G_S}^{(t)}))
\end{aligned}
$$

Except for the third term, the others are common training objective in diffusion models. Thus, we further derive the third item,

$$
\begin{aligned}
\mathbb{E}_{p_\phi}\log \frac{q(\mathbf{z}_{G_S}^{(T)}|\mathbf{z}_{G_S}^{(0)})}{p_\phi(\mathbf{z}_{G_S}^{(0)}|G)} &= \mathbb{E}_{p_\phi(\mathbf{z}_{G_S}^{(0)}|G),q(\mathbf{z}_S^{(t)}|\mathbf{z}_S^{(0)})}\log \frac{q(\mathbf{z}_{G_S}^{(t)}|\mathbf{z}_{G_S}^{(0)})p(\mathbf{z}_{G_S}^{(0)})}{p(\mathbf{z}_{G_S}^{(0)})p_\phi(\mathbf{z}_{G_S}^{(0)}|G)} \\
&\geq -D_{KL}(p(\mathbf{z}_{G_S}^{(0)})||p_\phi(\mathbf{z}_{G_S}^{(0)}|G))
\end{aligned}
$$

We can conclude that this term can be viewed as a regularization term, which constrains latent representation of graphs to prior distributions. Actually, our design in Equation 8 is corresponding to above regularization. We complete the proof and the details are provided in Appendix B.

## 5 THE IMPLEMENTATION OF SUBDIFF

Above analysis provides a discussion of the research motivations and key innovations of subgraph-level latent diffusion model. We will dissect the detailed implementations in this section.

## 5.1 ENCODING LATENT EMBEDDING VIA SUPERVISION

Given that the well-trained latent space should follow the property of input graphs, it should be learned with an explicit supervision. If the label values indicate the property evolution of an attribute, such as the biological activity of molecular, the $\mu_i$ of each sample then should follow a successive pattern representing by series of continuous values. If labels reveal pairwise inverse properties, the $\mu_i$ and $\mu_j$ regarding two samples with opposite properties should keep far away in the latent space.

In the embedding distribution space, we construct a mapping function between labels and the latent embedding space (described as a parameterized distribution), which enables achieving $\mu_i$ associated with $y_i$. As shown in Figure 1, we present two graphs with opposite solubility characteristics (*LogP*). Thus, the huge distance between $\mu_1$ and $\mu_2$ expresses their inverse property. Besides the six properties in our experiments with QM9 all indicate the evolution of attributes, we map the labels scale to $\mu_i \in [I, 2I]$. Details are provided in Appendix F.3.

Another note is that we directly constrain graph latent embedding rather than subgraph representation. Thus, the same subgraph in various graphs have different latent embedding encodered by SE-GNN. It is the context environment that determine the subgraph representation in different graphs.

## 5.2 TRAINING AND SAMPLING

With the proposed SE-GNN and HAS, we now present the training and sampling mechanism for SubDiff.

During training, the primary question is how to train the AE component and denoising module. Most practices of previous latent diffusion models have been proved that the two-stage training strategy usually leads to better performance. In our implementation, we inherit such two-stage training mechanism. First, we first train AE component to learn domain-special subgraph embeddings, and then train denoising module on subgraph latent embeddings. A detailed description of the training process is provided in Algorithm 1.

The HAS is our key technical contribution in the process of sampling. We design two sampling beginning $\mathbf{z}_{G_S}^{(T)}$ and $\mathbf{z}_{G_S}^{(0)}$ to unify unconditional and conditional generation. $\mathbf{z}_{G_S}^{(T)}$ is the beginning of traditional unconditional methods, which samples from the standard Gaussian distribution $\mathcal{N}(0, I)$ to the denoising module $\varepsilon_\theta$. Given desired property $y_k$, we sample $\mathbf{z}_{G_S}^{(0)}$ from distribution $\mathcal{N}(\mu_k, I)$, as the head of conditional generative process. We obtain the embedding of desired graph through the cycle of diffusion and denoising process. Then, we reconstruct the subgraph via well-trained $\mathcal{D}_\xi$ . A detailed description of the training process is provided in Algorithm 2. And, we also provide additional results about the setting of $\mu_i$ in Appdenix F.4.

## 5.3 3D MOLECULE GENERATION WITH SUBGRAPH

All neural networks used for the encoder, latent diffusion, and decoder, including $E_\theta$ and $D_\xi$, are implemented with SE-GNN. In SE-GNN, $L^1$ and $L^2$ are both basic backbones of GNNs to encode graph data, which is implemented by GIN (Xu et al., 2018). Recently, extensive works focus on generating molecules as 3D graphs (Xu et al., 2023; Hoogeboom et al., 2022). Actually, our experimental analysis is based on 3D molecular generation tasks. Each molecule is represented as point clouds $G = (\mathbf{x}, \mathbf{h}, \mathcal{E})$, where $\mathbf{x} \in \mathbb{R}^{N \times 3}$ is the atom coordinates matrix, $\mathbf{h} \in \mathbb{R}^{N \times d}$ is the node feature matrix, such as atomic type and charges, and $\mathcal{E}$ represents the connection information of nodes. The presence of coordinate information makes 3D graph different from traditional graph. For graph $G$ with $k$ subgraphs, we redefine $G = (\mathbf{x}_{G_S}, \mathbf{h}_{G_S}, \mathcal{E}^S)$ from the perspective of subgraph. $\mathbf{h}_{G_S} \in \mathbb{R}^{N \times d}$ denotes node feature matrix, and $\mathcal{E}_i^S$ represents the connection information between subgraphs. $\mathbf{x}_{G_S} \in \mathbb{R}^{k \times 3}$ represents subgraph-level coordinates matrix, where each element $\mathbf{x}_{G_S^i} = \frac{1}{|G_S^i|} \sum_{v \in G_S^i} \mathbf{x}_v$.

The reason of $\mathbf{h}_{G_S} \in \mathbb{R}^{N \times d}$ is our SE-GNN learns subgraph feature via node-level training. We aim to learn the latent embedding $\mathbf{z}_{G_S} = (\mathbf{z}_{G_S}^x, \mathbf{z}_{G_S}^h)$. The function $(\mathbf{z}_{G_S}^x, \mathbf{z}_{G_S}^h) = \mathcal{E}_\phi(\mathbf{x}_{G_S}, \mathbf{h}_{G_S})$ is E(3) equivariant if for any translation vector $\mathbf{t}$ and orthogonal matrix rotation $\mathbf{R}$, there exists:

$$\mathbf{R}\mathbf{z}_{G_S}^x + \mathbf{t}, \mathbf{z}_{G_S}^h = \mathcal{E}_\phi(\mathbf{R}\mathbf{x}_{G_S} + \mathbf{t}, \mathbf{h}_{G_S}) \tag{11}$$

Table 1: Results of atom stability, molecule stability, validity, and validity×uniqueness. A higher number indicates a better generationquality. The best results are in **bold** and the second best is underlined. The results of Data can be viewed as the upper bounds of all metrics.

| Metrics | QM9 | | | | DRUG | |
|---|---|---|---|---|---|---|
| | Atom Sta (%) | Mol Sta (%) | Valid (%) | Valid & Unique (%) | Atom Sta (%) | Valid (%) |
| Data | 99.0 | 95.2 | 97.7 | 97.7 | 86.5 | 99.9 |
| E-NF | 85.0 | 4.9 | 40.2 | 39.4 | - | - |
| G-Schnet | 95.7 | 68.1 | 85.5 | 80.3 | - | - |
| GDM | 97.0 | 63.2 | 89.1 | 87.4 | 75.0 | 90.8 |
| EDM | 98.7 | 82.0 | 91.9 | 90.7 | 81.3 | 92.6 |
| GEOLDM | 98.9 | 89.4 | 93.8 | **92.7** | 84.4 | 99.3 |
| SubDiff-VAE | 98.9 | 90.3 | 93.9 | 91.0 | 84.7 | 98.8 |
| NodeDiff | 97.9 | 89.1 | 92.9 | 90.9 | 84.2 | 98.4 |
| SubDiff | **98.9 ± 0.1** | **91.1 ± 0.8** | **94.2 ± 0.3** | 91.4 ± 0.4 | **85.3 ± 0.4** | **99.5 ± 0.1** |

Inspired by (Satorras et al., 2021), we slightly adjust the structure of SE-GNN to ensure E(3) equivariance. The feature and coordinate embedding are encoded by $L^1$ and $L^2$,

$$\mathbf{z}_{G_S}^h, \mathbf{z}_{G_S}^x = \left( L^1(\mathbf{h}_{G_S}, \mathcal{E}_{G_S}), \ \mathbf{x}_{G_S} \odot L^2(\mathbf{x}_{G_S}^{(0)}, \mathcal{E}^S) \right) \tag{12}$$

where each element of $\mathbf{x}_{G_S}^{(0)}$ is defined $\mathbf{x}_{G_S^i}^{(0)} = \lambda \sum_{j \neq i} ||\mathbf{x}_{G_S^j} - \mathbf{x}_{G_S^i}||^2$ and $\odot$ represents the element-wise multiplication. $\lambda$ is a hyperparameter set to 1. Note that $\mathbf{x}_{G_S}^{(0)}$ is calculated as the input of $L^2$, which is different from $\mathbf{x}_{G_S}$. This variant of SE-GNN can satisfy E(3) equivariance, we provide a formal proof in Appendix C.

# 6 EXPERIMENTS

## 6.1 EXPERIMENTAL SETTINGS

**Datasets.** QM9 (Ramakrishnan et al., 2014) is one of the most widely-used datasets for molecular-related tasks, which provides quantum chemical properties for a relevant, consistent, and comprehensive chemical space of small organic molecules. Following the common practice (Xu et al., 2023), we evaluate the conditional generation ability of SubDiff with 6 properties: polarizability $\alpha$, orbital energies $\varepsilon_{\text{HOMO}}$, $\varepsilon_{\text{LUMO}}$ and their gap $\Delta_\varepsilon$, Dipole moment $\mu$, and heat capacity $C_v$. GEOM-DRUG (Axelrod & Gomez-Bombarelli, 2022) is also one of the most popular datasets in molecule generation tasks. We follow the implementation (Hoogeboom et al., 2022) to select the 30 lowest energy conformations of each molecule for training.

**Baselines.** Our baselines are two-fold, i.e., diffusion-based generative models and others. (i) Equivariant Graph Diffusion Model (EDM) with its nonequivariant variant (GDM) (Hoogeboom et al., 2022) are representatFive studies on graph diffusion models. Geometric Latent Diffusion Model (GEOLDM) (Xu et al., 2023) is the first latent diffusion model for the molecular geometry domain. (ii) G-Schnet (Gebauer et al., 2019) and Equivariant Normalizing Flows (E-NF) (Garcia Satorras et al., 2021) are typical equivariant graph generative models, which are based on autoregressive and flow-based models respectively.

**Metrics.** We evaluate the ability of SubDiff in unconditional and conditional generation. (i) In unconditional generation task, we use *atom stability* (the proportion of atoms that have the right valency) and *molecule stability* (the proportion of generated molecules for which all atoms are stable), which are frequently used metrics in previous works (Hoogeboom et al., 2022) . Besides, we report *validity* and *uniqueness*, which are the proportion of valid and unique molecules among all the generated compounds (Xu et al., 2023). (ii) In conditional generation task, we test SubDiff on QM9 with 6 properties. We split QM9 and train a property prediction network $\omega$ (Hoogeboom et al., 2022). Given a desired property $C$, we draw samples from the generative models and input them into $\omega$ to

calculate their property values as $\widehat{C}$ (Garcia Satorras et al., 2021). We employ the *Mean Absolute Error (MAE)* between $C$ and $\widehat{C}$ to measure how close generated molecules to desired property.

## 6.2 MAIN RESULTS

Following previous graph generation works, we generate 10000 samples from each method to calculate the above metrics. The results of unconditional and conditional generation are reported in Table 1 and 2 respectively, and we have the following **Obs**ervations:

Table 2: Mean Absolute Error for molecular property prediction. The results of QM9 and Random can be viewed as lower and upper bounds of *MAE* on all properties.

| Property | $\alpha$ (Bohr$^3$) | $\Delta\varepsilon$ (meV) | $\varepsilon_{\text{HOMO}}$ (meV) | $\varepsilon_{\text{LUMO}}$ (meV) | $\mu$ (D) | $C_v$ ($\frac{\text{cal}}{\text{mol}}$K) |
|---|---|---|---|---|---|---|
| QM9 | 0.10 | 64 | 39 | 36 | 0.043 | 0.040 |
| Random | 9.01 | 1470 | 645 | 1457 | 1.616 | 6.857 |
| EDM | 9.01 | 655 | 356 | 584 | 1.111 | 1.101 |
| GEOLDM | 2.37 | 587 | 340 | 522 | 1.108 | 1.025 |
| SubDiff | 2.03 | 466 | 286 | 477 | 1.079 | 1.010 |

**Obs 1: In unconditional generation task, SubDiff is generally better than other generative models.** As shown in Table 1, we observe two main results, i.e. diffusion-based models are generally superior to traditional generation strategies and SubDiff is superior to prior diffusion-based frameworks. On the one hand, it demonstrates the dominance of diffusion mechanisms in generative models. On the other hand, SubDiff marginally outperforms current popular diffusion models. We argue that the reason is that the design principle based on subgraph play a crucial role in SubDiff.

**Obs 2: Subgraph-level latent diffusion model is superior to the node-level ones.** Compared with GEOLDM, a node-level latent diffusion model, the performance of molecules generated by SubDiff is more impressive. Except for *uniqueness* , SubDiff can achieve improvement than sub-optimal results on other metrics. The reason is that subgraph-level generation greatly simplifies the complexity of sampling process. Thus, all atoms inside subgraph are exactly with correct valency. This demonstrates that subgraph-level design can effectively improve the quality of generated samples.

**Obs 3: In conditional generation task, SubDiff can achieve outstanding performance in generating molecules with desired properties.** As shown in Table 2, SubDiff significantly outperforms baselines. The superior performance exhibits SubDiff's higher capacity for conditional generation and generate chemically realistic molecular geometries.

## 6.3 ABLATION STUDY

Our ablation studies aim to explore the effects of $\mathcal{L}_{recon}$ and subgraph-level generation. We design a variant of SubDiff with the same regularization term as VAE, named SubDiff-VAE. We also degenerate our model to a node-level latent diffusion model NodeDiff with the objective of $\mathcal{L}_{recon}$, which is totally different from GEOLDM. We keep SubDiff-VAE and NodeDiff with the same other configuration as SubDiff. Table 1 shows the performance comparison of SubDiff with its variants. We can observe the drop of NodeDiff and SubDiff-VAE performance. Specifically, SubDiff-VAE achieves many sub-optimal results and the performance of NodeDiff decreases more. Two conclusions can be reached, (i) subgraph-level study effectively promotes graph generation, (ii) the design of $\mathcal{L}_{recon}$ with supervised information not only unifies unconditional and conditional generation, but also can improve the quality of generated graphs. More discussion is provided in Appendix F.6.

## 7 CONCLUSION

In this work, we propose SubDiff, a novel latent diffusion model with subgraph as generative unit. To detour complex node-level dependencies in graph generations, we design a latent diffusion model by taking subgraph as the minimum unit. Besides, SubDiff also breaks the barrier between unconditional and conditional generative models via a Head Alterable Sampling strategy. We also provide theoretical analysis to prove that the training objective of SubDiff is the variational lower bound of the log-likelihood. Although subgraph-based graph generation achieves promising results, the exploration of subgraph-level interactions and generation for much larger graphs can be left as our future work.

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

## A  BACKGROUND AND RESEARCH MOTIVATION

### A.1  DIFFUSION MODEL

Diffusion models (DMs) learn data distributions by modeling the reverse of a diffusion process, which constructs two Markov chains. They respectively diffuse the data with predefined noise and reconstruct the desired samples from the noise.

In the forward diffusion chain, DMs gradually add Gaussian noise to data $x^{(t)}$ from raw data distribution $x_0 \sim q(x_0)$ for $t = 1, ..., T$, where $x_0$ denotes $G$ in graph diffusion model.

$$q(x^{(t)}|x^{(t-1)}) = \mathcal{N}(x^{(t)}; \sqrt{1-\beta_t}x^{(t-1)}, \beta_t I), \quad q(x^{(1:T)}|x^{(0)}) = \prod_{t=1}^{T} q(x^{(t)}|x^{(t-1)}) \quad (13)$$

where $\beta_t \in [0, 1]$ represents the variance of the Gaussian noise added at time step $t$. The final diffusion result is that samples $x^{(T)}$ can approximately converge to standard Gaussians, i.e., $q(x^{(T)}) \approx \mathcal{N}(0, 1)$. Note that this forward process $q$ is predefined without trainable parameters.

In the reverse denoising chain, DMs gradually reconstruct the original data from $x^{(T)}$ using $p_\theta$ parameterized by $\theta$.

$$p_\theta(x^{(t-1)}|x^{(t)}) = \mathcal{N}(x^{(t-1)}; \mu_\theta(x^{(t)}, t), \rho_t^2 I), \quad p_\theta(x^{(0:T)}) = p(x^{(T)}) \prod_{t=1}^{T} p_\theta(x^{(t-1)}|x^{(t)}) \quad (14)$$

where the $\mu_\theta$ is neural network, and the variance $\rho_t$ is also predefined. Since $q(x^{(1:T)}|x^{(0)})$ can be viewed as a fixed posterior, the learning process of DMs is to train $p_\theta$.

The variational lower bound of the likelihood of the data is given by

$$\log p(x^{(0)}) \geq \underbrace{\log p_\theta(x^{(0)}|x^{(1)})}_{\mathcal{L}_{reconstruct}} \underbrace{-D_{KL}(q(x^{(T)}|x^{(0)})||p(x^{(T)}))}_{\mathcal{L}_{prior}}$$
$$\underbrace{-\sum_{t=2}^{T} D_{KL}(q(x^{(t-1)}|x^{(t)}, x^{(0)})||p_\theta(x^{(t-1)}|x^{(t)}))}_{\mathcal{L}_{denoise}} \quad (15)$$

Thus, $p_\theta$ is trained to maximize the variational lower bound of the likelihood of the data:

$$\mathcal{L}_{vlb} = \mathcal{L}_{reconstruct} + \mathcal{L}_{prior} + \mathcal{L}_{denoise} \quad (16)$$

Further, (Song & Ermon, 2019; Ho et al., 2020) suggest a simple surrogate objective:

$$\mathcal{L}_{DM} = \mathbb{E}_{\varepsilon \sim N(0,I),t}[w(t)||\varepsilon - \varepsilon_\theta(x^{(t)}, t)||^2] \quad (17)$$

where $x_t = \alpha_t x_0 + \sigma_t \varepsilon$, with $\alpha_t = \sqrt{\prod_{s=1}^{t} (1-\beta_s)}$ and $\sigma_t = \sqrt{1-\alpha_t^2}$. We get $\varepsilon_\theta(x^{(t)}, t)$ from the parameterized method $\mu_\theta(x^{(t)}, t) := \frac{1}{\sqrt{1-\beta_t}}(x^{(t)} - \frac{\beta_t}{\sqrt{1-\alpha_t^2}}\varepsilon_\theta(x^{(t)}, t))$. And, the reweighting term is $w(t) := \frac{\beta_t^2}{2\rho_t^2(1-\beta_t)(1-\alpha_t^2)}$ while it has been proven empirically that setting it as 1 can promote the sampling quality. In the generation stage, we can draw samples with $\varepsilon_\theta$ by the iterative sampling:

$$x^{(t-1)} = \frac{1}{\sqrt{1-\beta_t}}(x^{(t)} - \frac{\beta_t}{\sqrt{1-\alpha_t^2}}\varepsilon_\theta(x^{(t)}, t)) + \rho_t \varepsilon \quad (18)$$

where $\varepsilon \sim \mathcal{N}(0, I)$.

## A.2 RESEARCH MOTIVATION

We investigate failure samples generated by prior works. A remarkable phenomenon we observe is most invalid samples stem from incorrect understanding of dependencies (connections) between nodes not by generated node set. This means that generative models always sampling correct nodes but fail to understand their complex dependencies. We provide empirical results in Table 3. For every generative models trained on QM9, we investigate 1000 failure samples. We calculate the proportion of these nodes that can be reconnected to form valid molecules (PRV). It verifies the motivation of our study.

Table 3: The proportion of failure samples that can be reconnected to generate valid molecules (PRV).

|         | PRV (%) |
|---------|---------|
| E-NF    | 70.3    |
| G-Schnet| 74.5    |
| GDM     | 80.8    |
| EDM     | 77.4    |
| GEOLDM  | 67.4    |

## B  THEOREM PROOF

**Theorem 1.** Let $\mathcal{L} := \mathcal{L}_{LDM} + \mathcal{L}_{AE}$. $\mathcal{L}$ is a variational lower bound to the log-likelihood , i.e., there is hold on for any $G$,

$$\mathbb{E}_{p_{data}}[\log p_{\phi,\xi,\theta}(G)] \geq -\mathcal{L} \tag{19}$$

*Proof.* We interchangeably use $z_{G_S}^{(0)}$ to denote original latent representation of subgraph set $z_{G_S}$. Then we have that,

$$\mathbb{E}_{p_{data}}[\log p_{\phi,\xi,\theta}(G)]$$

$$= \mathbb{E}_{p_\phi(\mathbf{z}_{G_S}|G)}\left[\log \frac{p_\xi(G|\mathbf{z}_{G_S}^{(0)})p(\mathbf{z}_{G_S}^{(0)})}{p_\phi(\mathbf{z}_{G_S}^{(0)}|G)}\right]$$

$$= \mathbb{E}_{p_\phi(\mathbf{z}_{G_S}|G)}\left[\log \int_{\mathbf{z}_{G_S}^{(1:T)}} \frac{p_\xi(G|\mathbf{z}_{G_S}^{(0)})p(\mathbf{z}_{G_S}^{(0:T)})}{p_\phi(\mathbf{z}_{G_S}^{(0)}|G)}\right]$$

$$= \mathbb{E}_{p_\phi(\mathbf{z}_{G_S}|G)}\left[\log \int_{\mathbf{z}_{G_S}^{(1:T)}} \frac{p_\xi(G|\mathbf{z}_{G_S}^{(0)})p(\mathbf{z}_{G_S}^{(0:T)})q(\mathbf{z}_S^{(1:T)}|\mathbf{z}_S^{(0)})}{p_\phi(\mathbf{z}_{G_S}^{(0)}|G)q(\mathbf{z}_S^{(1:T)}|\mathbf{z}_S^{(0)})}\right]$$

$$\geq \mathbb{E}_{p_\phi(\mathbf{z}_{G_S}^{(0)}|G),q(\mathbf{z}_S^{(1:T)}|\mathbf{z}_S^{(0)})}\left[\log \frac{p_\xi(G|\mathbf{z}_{G_S}^{(0)})p(\mathbf{z}_{G_S}^{(0:T)})}{p_\phi(\mathbf{z}_{G_S}^{(0)}|G)q(\mathbf{z}_S^{(1:T)}|\mathbf{z}_S^{(0)})}\right]$$

$$= \mathbb{E}_{p_\phi(\mathbf{z}_{G_S}^{(0)}|G),q(\mathbf{z}_S^{(1:T)}|\mathbf{z}_S^{(0)})}\left[\log p_\xi(G|\mathbf{z}_{G_S}^{(0)}) + \log \frac{p(\mathbf{z}_{G_S}^{(0:T)})}{p_\phi(\mathbf{z}_{G_S}^{(0)}|G)q(\mathbf{z}_S^{(1:T)}|\mathbf{z}_S^{(0)})}\right]$$

$$= \mathbb{E}_{p_\phi(\mathbf{z}_{G_S}^{(0)}|G)}[\log p_\xi(G|\mathbf{z}_{G_S}^{(0)})] + \mathbb{E}_{p_\phi(\mathbf{z}_{G_S}^{(0)}|G),q(\mathbf{z}_S^{(1:T)}|\mathbf{z}_S^{(0)})}\left[\log \frac{p(\mathbf{z}_{G_S}^{(0:T)})}{p_\phi(\mathbf{z}_{G_S}^{(0)}|G)q(\mathbf{z}_S^{(1:T)}|\mathbf{z}_S^{(0)})}\right]$$

$$= \mathbb{E}_{p_\phi(\mathbf{z}_{G_S}^{(0)}|G)}[\log p_\xi(G|\mathbf{z}_{G_S}^{(0)})] + \mathbb{E}_{p_\phi(\mathbf{z}_{G_S}^{(0)}|G),q(\mathbf{z}_S^{(1:T)}|\mathbf{z}_S^{(0)})}\left[\log \frac{p(\mathbf{z}_{G_S}^{(T)})\prod\limits_{t=1}^{T} p_\theta(\mathbf{z}_{G_S}^{(t-1)}|\mathbf{z}_{G_S}^{(t)})}{p_\phi(\mathbf{z}_{G_S}^{(0)}|G)\prod\limits_{t=1}^{T} q(\mathbf{z}_{G_S}^{(t)}|\mathbf{z}_{G_S}^{(t-1)},\mathbf{z}_S^{(0)})}\right]$$

$$= \mathbb{E}_{p_\phi(\mathbf{z}_{G_S}^{(0)}|G)}[\log p_\xi(G|\mathbf{z}_{G_S}^{(0)})] + \mathbb{E}_{p_\phi(\mathbf{z}_{G_S}^{(0)}|G),q(\mathbf{z}_S^{(1:T)}|\mathbf{z}_S^{(0)})}\left[\log \frac{p(\mathbf{z}_{G_S}^{(T)})}{p_\phi(\mathbf{z}_{G_S}^{(0)}|G)} + \log \prod\limits_{t=1}^{T} \frac{p_\theta(\mathbf{z}_{G_S}^{(t-1)}|\mathbf{z}_{G_S}^{(t)})}{q(\mathbf{z}_{G_S}^{(t)}|\mathbf{z}_{G_S}^{(t-1)},\mathbf{z}_S^{(0)})}\right]$$

$$= \mathbb{E}_{p_\phi(\mathbf{z}_{G_S}^{(0)}|G)}[\log p_\xi(G|\mathbf{z}_{G_S}^{(0)})] + \mathbb{E}_{p_\phi(\mathbf{z}_{G_S}^{(0)}|G),q(\mathbf{z}_S^{(1:T)}|\mathbf{z}_S^{(0)})}\log \frac{p(\mathbf{z}_{G_S}^{(T)})q(\mathbf{z}_{G_S}^{(T)}|\mathbf{z}_{G_S}^{(0)})}{q(\mathbf{z}_{G_S}^{(T)}|\mathbf{z}_{G_S}^{(0)})p_\phi(\mathbf{z}_{G_S}^{(0)}|G)}$$

$$+ \mathbb{E}_{p_\phi(\mathbf{z}_{G_S}^{(0)}|G),q(\mathbf{z}_S^{(1:T)}|\mathbf{z}_S^{(0)})}\left[\log \sum\limits_{t=1}^{T} \frac{p_\theta(\mathbf{z}_{G_S}^{(t-1)}|\mathbf{z}_{G_S}^{(t)})}{q(\mathbf{z}_{G_S}^{(t)}|\mathbf{z}_{G_S}^{(t-1)},\mathbf{z}_S^{(0)})}\right]$$

$$= \mathbb{E}_{p_\phi(\mathbf{z}_{G_S}^{(0)}|G)}[\log p_\xi(G|\mathbf{z}_{G_S}^{(0)})] - D_{KL}(q(\mathbf{z}_{G_S}^{(T)}|\mathbf{z}_{G_S}^{(0)})||p(\mathbf{z}_{G_S}^{(T)}))$$

$$+ \mathbb{E}_{p_\phi(\mathbf{z}_{G_S}^{(0)}|G),q(\mathbf{z}_S^{(1:T)}|\mathbf{z}_S^{(0)})}\log \frac{q(\mathbf{z}_{G_S}^{(T)}|\mathbf{z}_{G_S}^{(0)})}{p_\phi(\mathbf{z}_{G_S}^{(0)}|G)} - \sum\limits_{t=1}^{T} D_{KL}(q(\mathbf{z}_{G_S}^{(t)}|\mathbf{z}_{G_S}^{(t-1)},\mathbf{z}_S^{(0)})||p_\theta(\mathbf{z}_{G_S}^{(t-1)}|\mathbf{z}_{G_S}^{(t)}))$$

Except for the third term, the others are common training objective in diffusion models. Thus, we further derive the third item,

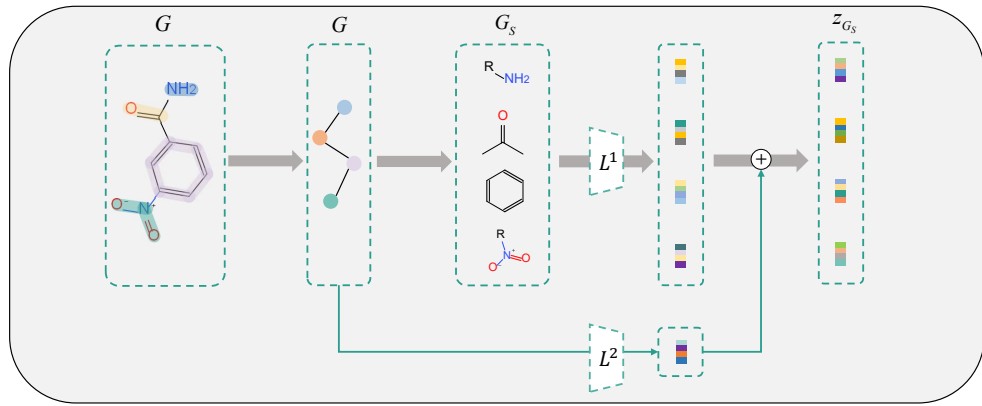

Figure 2: The architecture of SE-GNN. SE-GNN consists of the siamese component ($L^1$) and the information sharing component ($L^2$).

$$\mathbb{E}_{p_\phi(\mathbf{z}_{G_S}^{(0)}|G), q(\mathbf{z}_S^{(1:T)}|\mathbf{z}_S^{(0)})} \log \frac{q(\mathbf{z}_{G_S}^{(T)}|\mathbf{z}_{G_S}^{(0)})}{p_\phi(\mathbf{z}_{G_S}^{(0)}|G)}$$

$$= \mathbb{E}_{p_\phi(\mathbf{z}_{G_S}^{(0)}|G), q(\mathbf{z}_S^{(t)}|\mathbf{z}_S^{(0)})} \log \frac{q(\mathbf{z}_{G_S}^{(t)}|\mathbf{z}_{G_S}^{(0)})p(\mathbf{z}_{G_S}^{(0)})}{p(\mathbf{z}_{G_S}^{(0)})p_\phi(\mathbf{z}_{G_S}^{(0)}|G)}$$

$$= \mathbb{E}_{q(\mathbf{z}_S^{(t)}|\mathbf{z}_S^{(0)})} \log \frac{q(\mathbf{z}_{G_S}^{(t)}|\mathbf{z}_{G_S}^{(0)})}{p(\mathbf{z}_{G_S}^{(0)})} + \mathbb{E}_{p_\phi(\mathbf{z}_{G_S}^{(0)}|G)} \log \frac{p(\mathbf{z}_{G_S}^{(0)})}{p_\phi(\mathbf{z}_{G_S}^{(0)}|G)}$$

$$\geq D_{KL}(q(\mathbf{z}_{G_S}^{(t)}|\mathbf{z}_{G_S}^{(0)})||p(\mathbf{z}_{G_S}^{(0)})) - D_{KL}(p(\mathbf{z}_{G_S}^{(0)})||p_\phi(\mathbf{z}_{G_S}^{(0)}|G))$$

We can easily conclude that this term can be viewed as a regularization term, which constrain latent representation of graphs to prior distributions. Actually, our design in Equation 8 is corresponding to it. Therefore, we can obtain following conclusion,

$$\mathbb{E}_{p_{data}}[\log p_{\phi,\xi,\theta}(G)] \geq \underbrace{\mathbb{E}_{p_\phi(\mathbf{z}_{G_S}^{(0)}|G)}[\log p_\xi(G|\mathbf{z}_{G_S}^{(0)})]}_{\text{Reconstruction}} - \underbrace{D_{KL}(q(\mathbf{z}_{G_S}^{(T)}|\mathbf{z}_{G_S}^{(0)})||p(\mathbf{z}_{G_S}^{(T)}))}_{\text{Prior}}$$

$$- \underbrace{D_{KL}(p(\mathbf{z}_{G_S}^{(0)})||p_\phi(\mathbf{z}_{G_S}^{(0)}|G))}_{\text{Regularization}} - \underbrace{\sum_{t=1}^{T} D_{KL}(q(\mathbf{z}_{G_S}^{(t)}|\mathbf{z}_{G_S}^{(t-1)}, \mathbf{z}_S^{(0)})||p_\theta(\mathbf{z}_{G_S}^{(t-1)}|\mathbf{z}_{G_S}^{(t)}))}_{\text{Denoising}} = -ELBO$$

We finish the proof.

## C    EQUIVARIANCE PROOF

In this section we prove that our variant of SE-GNN is E(3) equivariant if for any translation vector $\mathbf{t}$ and orthogonal matrix rotation $\mathbf{R}$, there exists,

$$\mathbf{R}\mathbf{z}_{G_S}^x + \mathbf{t}, \mathbf{z}_{G_S}^h = \mathcal{E}_\phi(\mathbf{R}\mathbf{x}_{G_S} + \mathbf{t}, \mathbf{h}_{G_S}) \tag{20}$$

*Proof.* We analyze how a translation and rotation of the input coordinates propagates through our model. Thus, we focus on the input of $L^2$. Since $x^{(0)}$ is obtained by $(x^{(0)})_i = \lambda \sum_{j \neq i} ||x_{G_S^j} - x_{G_S^i}||^2$, $x^{(0)}$ is invariant. The reason is that the distance between two particles is invariant to translations.

$$||(\mathbf{R}\mathbf{x}_{G_S^i} + \mathbf{t}) - (\mathbf{R}\mathbf{x}_{G_S^j} + \mathbf{t}||^2 = ||\mathbf{R}\mathbf{x}_{G_S^i} - \mathbf{x}_{G_S^j})||^2 = ||(\mathbf{x}_{G_S^i} - \mathbf{x}_{G_S^j})||^2 \tag{21}$$

Therefore, we have

$$L^2(\mathbf{x}^{(0)}, \mathcal{E}^S) = L^2(\mathbf{R}\mathbf{x}^{(0)} + \mathbf{t}, \mathcal{E}^S) \tag{22}$$

We can further derive

$$
\begin{aligned}
(\mathbf{R}\mathbf{x}_{G_S} + \mathbf{t}) \odot L^2(\mathbf{R}\mathbf{x}_{G_S}^{(0)} + \mathbf{t}, E^S) &= (\mathbf{R}\mathbf{x}_{G_S} + \mathbf{t}) \odot L^2(\mathbf{x}_{G_S}^{(0)}, E^S) \\
&= \mathbf{R}(\mathbf{x}_{G_S} \odot L^2(\mathbf{x}_{G_S}^{(0)}, E^S)) + \mathbf{t} \\
&= \mathbf{R}\mathbf{z}_{G_S}^x + \mathbf{t}
\end{aligned}
\tag{23}
$$

Thus, we can conclude $\mathbf{R}\mathbf{z}_{G_S}^x + \mathbf{t}, \mathbf{z}_{G_S}^h = \mathcal{E}_\phi(\mathbf{R}\mathbf{x}_{G_S} + \mathbf{t}, \mathbf{h}_{G_S})$, where $\mathcal{E}_\phi$ is implemented by SE-GNN. We have proven that the variant of SE-GNN can satisfy E(3) equivariance.

## D  DETAILS OF IMPLEMENTATION

In this section, the subgraph extraction and dictionary generation processes are presented. We also provide detailed training and sampling process. [1]

### D.1  DETAILED MiCaM

MiCaM Geng et al. (2023) consists with two phases: **the merging-operation learning phase** and **the motif-dictionary construction phase**. This algorithm discovers the most common substructures based on their frequency of appearance in the molecule dictionary. We define each molecule in $\mathcal{D}$ is represented as a graph $G(\mathcal{V}, \mathcal{E})$, where the nodes $\mathcal{V}$ and edges $\mathcal{E}$ denote atoms and bonds respectively.

**The merging-operation learning phase** aims to learn the top $K$ most common patterns from dataset $\mathcal{D}$, where $K$ is a hyperparameter. For each $G(\mathcal{V}, \mathcal{E}) \in \mathcal{D}$, we use a merging graph $G_\mathcal{M}(\mathcal{V}_\mathcal{M}, \mathcal{E}_\mathcal{M})$ to track the merging status, i.e., to represent the fragments and their connections. In $G_\mathcal{M}(\mathcal{V}_\mathcal{M}, \mathcal{E}_\mathcal{M})$, each node $\mathcal{F} \in \mathcal{V}_\mathcal{M}$ represents a fragment (either an atom or a subgraph) of the molecule, and the edges in $\mathcal{E}_\mathcal{M}$ indicate whether two fragments are connected with each other. MiCaM first initializes each merging graph from the molecule graph by treating each atom as a single fragment and inheriting the bond connections from $G$, i.e., $G_\mathcal{M}^{(0)}(\mathcal{V}_\mathcal{M}^{(0)}, \mathcal{E}_\mathcal{M}^{(0)}) = G(\mathcal{V}, \mathcal{E})$. MiCaM defines an operation $\oplus$ to create a new fragment $\mathcal{F}_{ij} = \mathcal{F}_i \oplus \mathcal{F}_j$ by merging two fragments $\mathcal{F}_i$ and $\mathcal{F}_j$ together. The newly obtained $\mathcal{F}_{ij}$ contains all nodes and edges from $\mathcal{F}_i, \mathcal{F}_j$, as well as all edges between them. Then, MiCaM iteratively updates the merging graphs to learn merging operations. In the merging graph $G_\mathcal{M}^{(k)}(\mathcal{V}_\mathcal{M}^{(k)}, \mathcal{E}_\mathcal{M}^{(k)})$ at the $k^{\text{th}}$ iteration ($k = 0, ..., K-1$), each edge represents a pair of fragments, $(\mathcal{F}_i, \mathcal{F}_j)$, that are adjacent in the molecule. It also gives out a new fragment $\mathcal{F}_{ij} = \mathcal{F}_i \oplus \mathcal{F}_j$. MiCaM traverses all edges $(\mathcal{F}_i, \mathcal{F}_j) \in \mathcal{E}_\mathcal{M}^{(k)}$ in all merging graphs $G_\mathcal{M}^{(k)}$ to count the frequency of $\mathcal{F}_{ij} = \mathcal{F}_i \oplus \mathcal{F}_j$, and denote the most frequent $\mathcal{F}_{ij}$ as $\mathcal{M}^{(k)}$. Consequently, the $k$-th merging operation is defined as: if $\mathcal{F}_i \oplus \mathcal{F}_j == \mathcal{M}^{(k)}$, then merge $\mathcal{F}_i$ and $\mathcal{F}_j$ together. MiCaM applies the merging operation on all merging graphs to update them into $G_\mathcal{M}^{(k+1)}(\mathcal{V}_\mathcal{M}^{(k+1)}, \mathcal{E}_\mathcal{M}^{(k+1)})$. We repeat such a process for $K$ iterations to obtain a merging operation sequence $\{\mathcal{M}^{(k)}\}_{k=0}^{K-1}$.

**The motif-dictionary construction phase** repeats this process for a pre-defined number of steps and collect the fragments to build a motif dictionary. For each molecule $G(\mathcal{V}, \mathcal{E}) \in \mathcal{D}$, MiCaM applies the merging operations sequentially to obtain the ultimate merging graph $G_\mathcal{M}(\mathcal{V}_\mathcal{M}, \mathcal{E}_\mathcal{M}) = G_\mathcal{M}^{(0)}(\mathcal{V}_\mathcal{M}^{(K)}, \mathcal{E}_\mathcal{M}^{(K)})$. MiCaM then disconnects all edges between different fragments and add the symbols $*$ to the disconnected positions. The fragments with $*$ symbols are connection-aware, and we denote the connection-aware version of a fragment $F$ as $\mathcal{F}^*$. The motif vocabulary is the collection of all such connection-aware motifs: $Dictionary = \cup_{G_\mathcal{M}(\mathcal{V}_\mathcal{M}, \mathcal{E}_\mathcal{M}) \in \mathcal{D}} \{\mathcal{F}^* : \mathcal{F} \in \mathcal{V}_\mathcal{M}\}$.

---

[1] The code of SubDiff is available at https://anonymous.4open.science/r/SubDiff-6F90.

## D.2 Detailed Training and Sampling

---

**Algorithm 1:** Training Algorithm of SubDiff

---

**Input:** graph data $G = (\mathcal{X}, \mathcal{E})$
**Output:** encoder network $\mathcal{E}_\phi$, decoder network $\mathcal{D}_\xi$, denoising network $\varepsilon_\theta$
1 **First Stage: Extract Subgraphs**
2 $G_S \leftarrow G$
3 **Second Stage: Autoencoder Training**
4 **while** $\mathcal{E}_\phi$ *and* $\mathcal{D}_\xi$ *have not converged* **do**
5 $\quad \mathbf{z}_{G_S} \leftarrow \mathcal{E}_\phi(G_S)$
6 $\quad \widetilde{G_S} \leftarrow \mathcal{D}_\xi(\mathbf{z}_{G_S})$
7 $\quad \mathcal{L}_{AE} = \mathcal{L}_{recon} + \mathcal{L}_{reg}$
8 $\quad \phi, \xi \leftarrow optimizer(\mathcal{L}_{AE}; \phi, \xi)$
9 **end**
10 **Third Stage: Latent Diffusion Model Training**
11 **while** $\varepsilon_\theta$ *have not converged* **do**
12 $\quad \mathbf{z}_{G_S}^{(0)} \leftarrow E_\phi(G_S)$
13 $\quad t \sim \mathrm{U}(0, T), \quad \varepsilon \sim \mathcal{N}(0, I)$
14 $\quad \mathbf{z}_{G_S}^{(t)} = \alpha_t \mathbf{z}_{G_S}^{(0)} + \sigma_t \varepsilon$
15 $\quad \mathcal{L}_{LDM} = \mathbb{E}_{\varepsilon \sim \mathcal{N}(0,I),t}[w(t)||\varepsilon - \varepsilon_\theta(\mathbf{x}^{(t)}, t)||^2]$
16 $\quad \theta \leftarrow optimizer(\mathcal{L}_{LDM}; \theta)$
17 **end**
**Result:** $\mathcal{E}_\phi$, $\mathcal{D}_\xi$ and $\varepsilon_\theta$

---

---

**Algorithm 2:** Sampling Algorithm of SubDiff

---

**Input:** encoder network $\mathcal{E}_\phi$, decoder network $\mathcal{D}_\xi$, denoising network $\varepsilon_\theta$ and condition $C$
**Output:** generative graph $G$
1 **if** $C$ *is None* **then**
2 $\quad \varepsilon \sim \mathcal{N}(0, I)$
3 **else**
4 $\quad y_k \leftarrow C$
5 $\quad \widehat{\mathbf{z}_{G_S}^{(0)}} \sim \mathcal{N}(\mu_k, I)$
6 $\quad \varepsilon := \mathbf{z}_{G_S}^{(T)} \leftarrow \widehat{\mathbf{z}_{G_S}^{(0)}}$
7 **end**
8 **for** $t \leftarrow T, T-1, ..., 1$ **do**
9 $\quad \mathbf{z}_{G_S}^{(t-1)} = \frac{1}{\sqrt{1-\beta_t}}(\mathbf{z}_{G_S}^{(t)} - \frac{\beta_t}{\sqrt{1-\alpha_t^2}}\varepsilon_\theta(\mathbf{z}_{G_S}^{(t)}, t)) + \rho_t \varepsilon$
10 **end**
**Result:** $G_S \sim \mathcal{D}_\xi(\mathbf{z}_{G_S})$

---

## E More Related Works

As a classic discrete graph diffusion model on molecule generation tasks, MDM(Huang et al., 2023) give me some inspiration. Diffusion-based methods for generating molecules always suffer from poor performance with large molecules and lack diversity. MDM addresses these challenges by incorporating interatomic relations and using dual equivariant encoders to capture interatomic forces of different strengths. It also introduces a distributional controlling variable to improve exploration and increase generation diversity. Extensive experiments demonstrate that MDM outperforms existing methods for both unconditional and conditional molecule generation tasks.

Besides, as one of early works on graph latent diffusion models, NVDiff (Chen et al., 2022) generates novel and realistic graphs by taking the VGAE structure and uses SGM as its prior for latent node vectors. More important, this work proven that the latent graph diffusion model has a proper lower-bound of the graph likelihood.

# F SUPPLEMENTARY EXPERIMENTS

## F.1 DETAILS OF THE DATASETS

**QM9** (Ramakrishnan et al., 2014) is a widely used benchmark for the prediction of physical properties of molecules in equilibrium. It consists of around 130k small organic molecules with up to 9 heavy atoms (C, O, N, and F). The properties are computed using density functional theory (DFT) calculations. It provides quantum chemical properties for a relevant, consistent, and comprehensive chemical space of small organic molecules.

**GEOM-DRUG** (Axelrod & Gomez-Bombarelli, 2022) is a larger scale dataset of molecular conformers. It features 430000 molecules with up to 181 atoms and 44.4 atoms on average. For each molecule, many conformers are given along with their energy. From this dataset, we retain the 30 lowest energy conformations for each molecule. The models learn to generatethe 3D positions and atom types of these molecules. One of the most notable features of GEOM-DRUG is that molecules in this dataset are bigger and have more complex structures than QM9.

## F.2 DETAILS OF THE BASELINES

**EDM** and its nonequivariant variant **GDM** (Hoogeboom et al., 2022) are representative studies on graph diffusion models. EDM is an E(3) equivariant diffusion modelfor molecule generation in 3D, which scales better and can generate valid conformations while explicitly modeling hydrogen atoms. GDM is the nonequivariant variant of EDM, which is design as an ablation architecture with non-equivariant component.

**GEOLDM** (Xu et al., 2023) is the first latent diffusion model for the molecular geometry domain. To address the limitations caused by prior models operating directly on high-dimensional, multi-modal atom features, GEOLDM learns diffusion models over a continuous, lower-dimensional latent space. By building point-structured latent codes with both invariant scalars and equivariant tensors, GEOLDM is able to effectively learn latent representations while maintaining roto-translational equivariance.

**G-Schnet** (Gebauer et al., 2019) is a autoregressive neural network, which is proposed for the generation of 3d point sets incorporating the constraints of Euclidean space and rotational invariance of the atom distribution as prior knowledge. It both incorporates the constraints of euclidean space and the spatial invariances of the targeted geometries. This is achieved by determining the next position using distances to previously placed points, resulting in an equivariant conditional probability distribution.

**E-NF** (Garcia Satorras et al., 2021) is a generative model equivariant to Euclidean symmetries. E-NF is continous-time normalizing flow that utilize an EGNN with improved stability as parametrization.

## F.3 DETAILS OF LATENT EMBEDDING SPACE

In practices, we test the ability of SubDiff conditional generation with 6 properties: polarizability $\alpha$, orbital energies $\varepsilon_{\text{HOMO}}$, $\varepsilon_{\text{LUMO}}$ and their gap $\Delta_\varepsilon$, Dipole moment $\mu$, and heat capacity $C_v$. The label embedding space for all these 6 properties is successive. For example, the dipole moment $\mu$ of a molecule can be used to predict its polarity. A larger dipole moment of the molecule results in a larger polarity. Based on the value of dipole moment, we map the mean $\mu_i$ of each element to $[I, 2I]$. Detailed setting is provided in Table 4.

## F.4 THE SETTING OF $\mu_i$

In our insights, we should maintain the semantic consistency between the facts of graph properties and latent representation space, and graph properties are considered as the supervision signal in our implementation. Thus, the assumption of the latent embeddings actually comes from the physical facts.

Specifically, the setting of $\mu_i$ is to control the 'distance' between sample representations with property value $y_i$ . In our implementations, since the six properties we use are all evolving continuous variables,

Table 4: The setting of mean $\mu_i$ .

| Property | Description | Unit | Label Relation | $\mu_i$ |
|---|---|---|---|---|
| $\alpha$ | Isotropic polarizability | $a_0{}^3$ | Continuous | $[I, 2I]$ |
| $\varepsilon_{\text{HOMO}}$ | Highest occupied molecular orbital energy | eV | Continuous | $[I, 2I]$ |
| $\varepsilon_{\text{LUMO}}$ | Lowest unoccupied molecular orbital energy | eV | Continuous | $[I, 2I]$ |
| $\Delta_\varepsilon$ | Gap between $\varepsilon_{\text{HOMO}}$ and $\varepsilon_{\text{LUMO}}$ | eV | Continuous | $[I, 2I]$ |
| $\mu$ | Dipole moment | $D$ | Continuous | $[I, 2I]$ |
| $C_v$ | Heat capavity at 298.15K | $\frac{\text{cal}}{\text{mol}}$K | Continuous | $[I, 2I]$ |

Table 5: Results of the setting of $\mu_i$.

| | Atom Sta (%) | Mol Sta (%) | Valid (%) | Valid & Unique (%) |
|---|---|---|---|---|
| SubDiff($[0, I]$) | 97.8 ± 1.2 | 89.9 ± 1.6 | 91.4 ± 1.2 | 89.2 ± 1.9 |
| SubDiff($[I, 2I]$) | 98.9 ± 0.1 | 91.1 ± 0.8 | 94.2 ± 0.3 | 91.4 ± 0.4 |

we map the range of property values to $k = [1, 2]$ and constrain the corresponding sample latent representation by $\mu_i = k\mathbf{I} \in [\mathbf{I}, 2\mathbf{I}]$ where $\mathbf{I}$ is the unit matrix.

I guess that you are confused about why $\mu_i \in [I, 2I]$ is used as the priori instead of $[0, I]$. We experimentally explored this problem in the early stages of our research, and now show empirical results on QM9 as below. In most evaluation metrics, different settings can achieve similar overall performance, but $[0, I]$'s design is obviously more unstable.

### F.5 Extracted Subgraph Dictionary

As described in Section 4.1, we present that the veracity of each subgraph and the sufficiency of subgraph dictionary should be primarily guaranteed in the stage of subgraph discovery. The veracity means that the extracted subgraphs should be domain-meaningful, while the sufficiency means that we should extract all domain-related subgraphs to form subgraph dictionary. Thus, we provide visual results of extracted dictionary. Figure 3 show the 3D subgraph dictionary extracted by MiCaM. We obtain this dictionary according to the configuration of (Geng et al., 2023). Except for extracted subgraphs, five atoms types (H, C, N, O, F) and integer-valued atom charges as atomic features are also presented in the dictionary on the generation tasks based on QM9. For DRUG, we only add five atom types to subgraphs dictionary.

### F.6 Ablation Study

The target of our ablation study is to explore the effects $\mathcal{L}_{recon}$ and subgraph-level generation. Thus, we design two variants of SubDiff to achieve our exploration. Table 1 shows the performance comparison of SubDiff with its variants.

**SubDiff-VAE** is a variant of SubDiff obtained by replacing $\mathcal{L}_{recon}$ with the same regularization term as VAE. We observed the performance drop of SubDiff-VAE compared with SubDiff in all metrics. This means that $\mathcal{L}_{recon}$ contributes to generating excellent graphs. However, it is worth noting that the SubDiff-VAE still obtains some sub-optimal results in some metrics. We argue that the design of $\mathcal{L}_{recon}$ mainly play a vital role in unifying unconditional and conditional generative schemes. It does not cause significant performance degradation on unconditional generative tasks.

**NodeDiff** is a variant of SubDiff achieved by degenerating it to a node-level latent diffusion model. It is with the objective of $\mathcal{L}_{recon}$, which is totally different from GEOLDM. Compared with SubDiff-VAE, the performance of NodeDiff dropped more sharply. We argue that SubDiff is particularly outstanding in the use of subgraph latent embedding. Subgraph latent diffusion can achieve great improvement over node-based models. However, GEOLDM is stronger than NodeDiff on many metrics. We think NodeDiff with $\mathcal{L}_{recon}$ not appropriate. Subgraph have the characteristic of directly determining the properties of graph, but node don't have this characteristic. Thus, the design of $\mathcal{L}_{recon}$ with supervision is conducive to obtaining excellent subgraph latent embedding, but does not contribute to good node-level latent embedding.

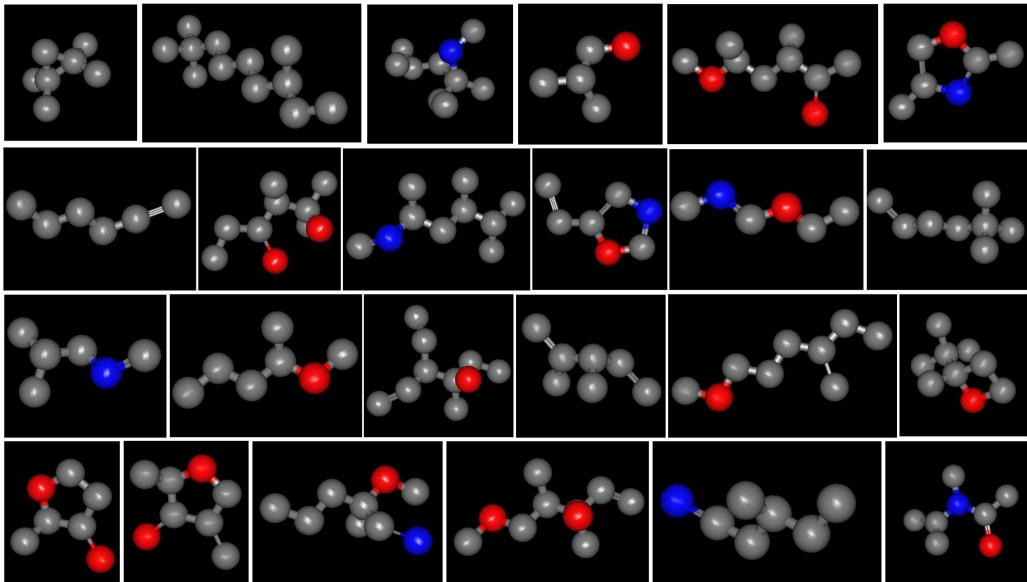

Figure 3: The subgraph dictionary extracted by MiCaM on QM9.

The above discussion drives two conclusions, (i) subgraph-level study effectively promotes graph generation, (ii) the design of $\mathcal{L}_{recon}$ with supervised information not only unifies unconditional and conditional generation, but also can improve the quality of generated graphs.

### F.7 MORE VISUALIZATION RESULTS

In this section, we provide more visualizations of molecules generated from SubDiff. We mainly analyze generated molecules with given conditions. We visualize the samples whose $\Delta_\varepsilon$ are desired labels (eV). Figure 4 shows some samples generated by SubDiff with given condition $C$ and their real labels $\widehat{C}$. We observe that most generated molecules are small size graphs. We argue that there are two main reasons, i.e., the limitations of datasets and the preference to generate models.

For the limitations of datasets, we think more datasets with complex graph structure should be proposed and explored. For the preference to generate models, the scale of generated samples perhaps should also be considered in the model design process. These two issues will also be further studied in our future works.

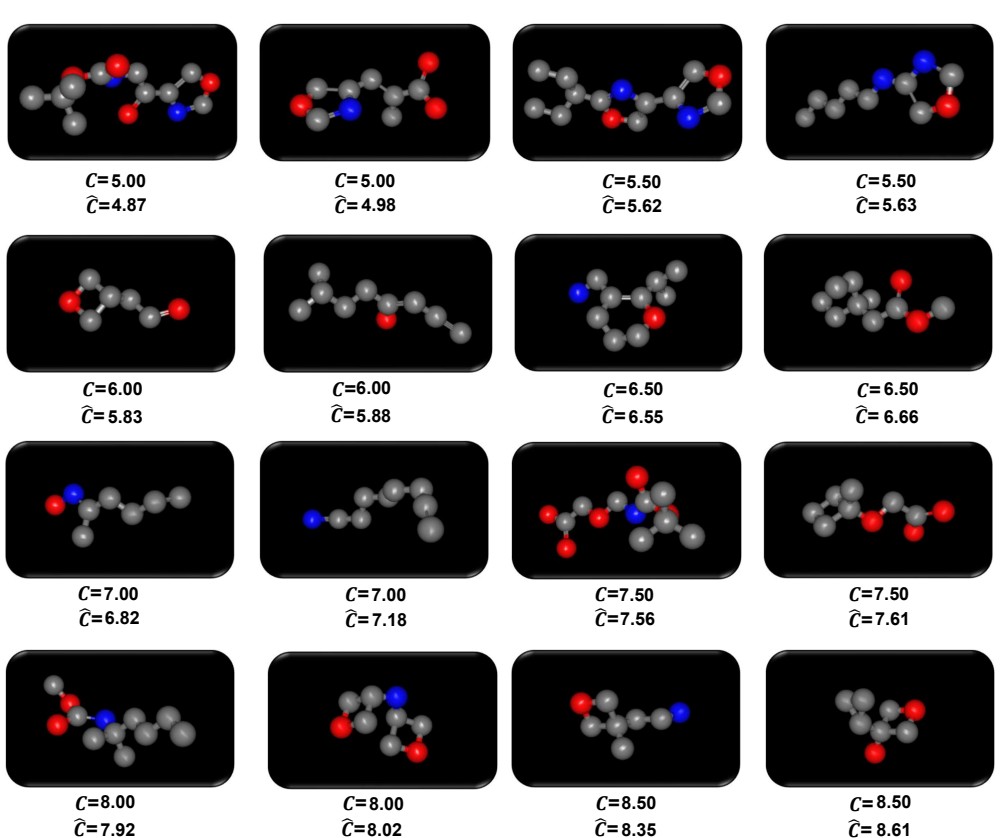

Figure 4: The samples generated by SubDiff with given condition $C$ and their real labels $\widehat{C}$ (eV).

