# OpenReview forum: "SubDiff: Subgraph Latent Diffusion Model"
_ICLR.cc/2024/Conference — ICLR 2024 Conference Withdrawn Submission_

### Official Review · Reviewer_tXm3 · 2023-10-30

**Soundness:** 3 good
**Presentation:** 2 fair
**Contribution:** 3 good
**Rating:** 5
**Confidence:** 4

**Summary:**

This paper proposes a subgraph latent diffusion model to embed the subgraph into the latent space. The explicit supervision used in the subgraph latent diffusion model helps to embed the label information in latent space. A novel subgraph equivariant GNN is also raised to extract the graph representation. A sampling method HES is also devised to unify conditional and unconditional generative learning.

**Strengths:**

- The paper considers the subgraphs as minimum units instead of separate nodes, which makes sense and has the potential to enhance the substructure perception of GNNs.
- Embeding the condition information to the latent space sounds interesting.
- The paper proposes a simple method to unify the conditional and unconditional generation via setting different starting Gaussian noise.

**Weaknesses:**

- The paper claims that they propose a new frequency-based subgraph extractor. However, the method actually used is MiCaM, proposed by (Geng et al., 2023).
- The assumption of the latent embeddings is strong (sec 5.1): the condition must be numerical from 0 to 1 and comparable.

- The presentation of this paper is not unclear. It misses many important details in the main text, such as model architecture(see minor concerns), and sampling process.
- There exist many approaches that can be used to extract the subgraph, such as BRICS. The ablation study can be added to support the choice of MiCaM.


Minor concerns:
- The explanation of $E_{\theta}$ and $D_{\xi }$ in Eq (5) are missing.
- The explanation of “pooling” in Eq (7) is missing.
- In Eq 12, the specific forms of  $L^2$ and $L^2$ are not given. In this case, how to calculate the element-wise multiplication between $x_{G_s}$ and $L^2$?

 While I think the subgraph diffusion is a promising idea, the presentation of the method and experiments require a substantial amount of work and are not ready for ICLR24.

**Questions:**

- What is the meaning of Proposition 1? From my understanding, if we get an unconditional generative model, the model can be easily extended to a conditional version. E.g. EDM.
- The input of the denoising network in Alg. 1(training process) is x while z in Alg. (sampling)? Why?

---

> ### Author Response · Authors · 2023-11-14
> **Reply to Reviewer tXm3 (1)**
>
> Dear Reviewer tXm3,
>
> Thanks for your careful reading and feedback. We are really appreciated and well encouraged for your recognition of our work.
>
> **W1. Frequency-based subgraph extractor.**
>
> Actually, we haven’t claimed that we propose a new frequency-based subgraph extractor, but just implement the subgraph extractor by MiCaM[1] (have cited in manuscript). The aim of our research is to achieve unified and interpretable graph diffusion generation but not the subgraph extraction. Therefore, our extractor is directly realized by existing support method.
>
> **W2. The setting of $ {\mu _i}$.**
>
> In our insights, we should maintain the semantic consistency between the facts of graph properties and latent representation space, and graph properties are considered as the supervision signal in our implementation. Thus, the assumption of the latent embeddings actually comes from the physical facts.
>
> Specifically, the setting of $ {\mu _i}$ is to control the ‘distance’ between sample representations with property value $y_i$ . In our implementations, since the six properties we use are all evolving continuous variables, we map the range of property values to $k=[1,2]$ and constrain the corresponding sample latent representation by ${\mu _i} = k\mathbf{I} \in [\mathbf{I},2\mathbf{I}]$ where $\mathbf{I}$ is the unit matrix.
>
> I guess that you are confused about why $ {\mu _i} \in [I,2I]$ is used as the priori instead of $[0,I]$. We experimentally explored this problem in the early stages of our research, and now show empirical results on QM9 as below. In most evaluation metrics, different settings can achieve similar overall performance, but $[0,I]$'s design is obviously more unstable.
>
> |      | Atom Sta (% ) | Mol Sta (% ) |  Valid (% )  | Valid & Unique (% )|
> |   :--:   | :--:   | :--:   |  :--: | :--: |
> | SubDiff($[0,I]$) |  97.8 ± 1.2   |   89.9 ± 1.6  |  91.4 ± 1.2  |  89.2 ± 1.9  |
> | SubDiff($[I,2I]$) |  98.9 ± 0.1   |   91.1 ± 0.8  |  94.2 ± 0.3  |  91.4 ± 0.4  |
>
> We make the following hypothesis based on above results. Note that the goal of noising is $z_{{G_S}}^{(T)} \sim N(0,I)$. If $ [0,I]$ is set, the noising process of all samples from $ z_{{G_S}}^{(0)}$ to $z_{{G_S}}^{(T)}$ is equivalent to trapping all samples close to those samples with ${\mu _i} = 0$. Such process makes the diffusion (noise adding) process becomes biased and will greatly reduce the diversity of generated samples. Therefore, we directly preset ${\mu _i} \in [I,2I]$ as a priori.
>
> However, the above analysis is just derived from the hypotheses of empirical results. Thus it still has not discussed in our manuscript. Maybe it will be further studied in the future with both empirical and theoretical support.
>
> **W3. Sampling process.**
>
> **Insights and solution overview. ** The graph diffusion process aims to construct distribution from training samples, and generate new graphs by sampling from constructed distribution. In our work, we establish two distributions as the separated sampling beginning heads to unify the conditional and unconditional generation task. We argue that different distributions can be controlled by different sampling beginnings, which result in various generated contents. Therefore, we let the beginning distribution of conditional generation with supervision ($ H_C $) while unconditional generation sample from pure Gaussian distribution ($ H_U $).
>
> To be specific, we propose the ‘Head Alterable Sampling’ strategy, where ‘Head Alterable’ refers to the adjustable beginning (position), where it has been described in Sec. 5.2 and Fig. 1(c). We further show their respective generation (sampling) paths:
>
> ${H_U}:z_{{G_S}}^{(T)} -  -  > \hat z_{{G_S}}^{(0)} -  -  > G$
>
> ${H_C}:z_{{G_S}}^{(0)} -  -  > z_{{G_S}}^{(T)} -  -  > \hat z_{{G_S}}^{(0)} -  -  > G$
>
>
> where $z_{{G_S}}^{(0)}$ ~ $N({\mu _i},I)$ and ${\mu _i}$ is pre-designed based on the characteristics of our focused properties as shown in Table 4.
>
>
> **W4. Ablation Study.**
>
> Since the subgraph extractor is a sub-core part, we take MiCaM as the subgraph extractor, which is able to guarantee the veracity and sufficiency of extracted subgraphs. In fact, the authors of MiCaM [1] have shown a performance comparison with BRICS [2], and MiCaM obtained better performance.
>
>
> **W5. Minor concerns.**
>
> All neural networks used for the encoder, latent diffusion, and decoder, including ${E_\theta }$ and $ {D_\xi }$, are implemented with SE-GNN. In SE-GNN, ${L^1}$ and ${L^2}$ are both basic backbones of GNNs to encode graph data, which is implemented by GIN. $ pooling $ is MEAN operation in our work.

---

> ### Author Response · Authors · 2023-11-14
> **Reply to Reviewer tXm3 (2)**
>
> **Q1. Proposition 1.**
>
> Proposition 1 indicates that we support the perspective that the generative model inductively learns the distribution space of training samples, and samples from this space to achieve specific generative tasks [3, 4]. Specifically, unconditional generation (Un-G) focuses on the wideness of the sampling space to enable diversity, while conditional generation (Con-G) focuses on the bounds of specific properties in the sampling space. In essence, Con-G sample space is a certain subset (cluster) of Un-G space.
>
> Actually, for almost all previous graph generative models [5, 6, 7], the unconditional model can be extended to conditional version with additional training. In contrast, our work is to unify the separated training schemes on conditional/unconditional generation into once training paradigm. The superiority of such once training can fully exploit the existing samples for comprehensive understanding, which not only enhances the quality of conditional generation but also improves the efficiency of conditional-unconditional learning.
>
> **Q2. Writing issue in Alg. 1.**
>
> Thanks again for your careful review! The $x^{(t)}$ in the training phase should be written as $z_{{G_S}}^{(t)}$. We will thoroughly check the spelling and grammar issues in this manuscript.
>
> Your comments are very professional and insightful, which give us a deeper understanding to improve our work. Many of your observations also inspired our future work. We will incorporate these results and additional discussions in our revised submission. Thanks again! Looking forward to your reply!
>
>
> **References:**
>
> [1] De novo molecular generation via connection-aware motif mining, ICLR 2023.
>
> [2] On the art of compiling and using ’drug-like’ chemical fragment spaces, 2008.
>
> [3] Elucidating the Design Space of Diffusion-Based Generative Models, NeurIPS 2022.
>
> [4] Subspace diffusion generative models, ECCV 2022.
>
> [5] Equivariant diffusion for molecule generation in 3d, ICML 2022.
>
> [6] Geometric Latent Diffusion Models for 3D Molecule Generation, ICML 2023.
>
> [7] Generative diffusion models on graphs: Methods and applications, 2023.

---

> > ### Author Response · Authors · 2023-11-20
> > **Reply to Reviewer tXm3 (3)**
> >
> > Dear Reviwer tXm3,
> >
> > Thank you once again for dedicating your valuable time to reviewing our work. We have already submitted our response six days ago and sincerely hope it can address your concerns. We also note that that there are only 48 hours remaining for the discussion phase. Thus, we would like to know whether our response adequately addresses your questions, and if there are any additional inquiries you may have.
> >
> > We sincerely look forward to your response.
> >
> > Thanks!

---

> > > ### Comment · Reviewer_tXm3 · 2023-11-23
> > >
> > > Thanks to the authors for the reply!
> > > In the last paragraph of the Introduction: "a frequency-based subgraph extractor and a novel subgraph equivariant framework is
> > > propsoed to encode subgraph latent embedding." Is this sentence correct?

---

> > > > ### Author Response · Authors · 2023-11-23
> > > > **Thanks for your follow-up**
> > > >
> > > > Dear Review tXm3,
> > > >
> > > > Thanks for your your follow-up. We have updated our manuscript, and your comments have been very helpful to us. As a first attempt at the Subgraph Latent Diffusion Model, our approach is significantly different from existing methods and shows remarkable performance improvements. Therefore, we sincerely hope that the reviewers can provide an opportunity to consider our work in the next round.
> > > >
> > > > Thanks again!

---

> > > > > ### Comment · Reviewer_tXm3 · 2023-11-23
> > > > >
> > > > > Thanks for the response!
> > > > > I maintain my score.

---

### Official Review · Reviewer_C6xG · 2023-10-31

**Soundness:** 3 good
**Presentation:** 3 good
**Contribution:** 3 good
**Rating:** 6
**Confidence:** 4

**Summary:**

The paper proposes a subgraph diffusion model that learns to treat the subgraph as the basic component of the diffusing object. To do so, it overcomes several design challenges, which are demonstrated in the paper thoroughly.

**Strengths:**

The proposed method is technically solid. (1) it has shown that treating subgraphs as the latent variables can also maintain a lower bound of the graph likelihood; (2) it tackles the Equivariant problem when treating subgraphs as diffusion object; (3) experiment result has shown promising result of the propose method

**Weaknesses:**

(1) the motivation that drives such an approach may not be sufficient -- it's not very convincing that subgraph-level diffusion will address the problem "graph generative models generate not only the features of each node but also the complex semantic association between nodes."

(2) The claim that the model unifies condition and unconditional generation seems to be irrelevant to the subgraph diffusion. It's not sure why these two components are proposed in one submission

(3) Related works missing -- there is a previous work that has proven that the latent graph diffusion model has a proper lower-bound of the graph likelihood [1].

[1] Chen, Xiaohui, et al. "Nvdiff: Graph generation through the diffusion of node vectors." arXiv preprint arXiv:2211.10794 (2022).

**Questions:**

See weakness

---

> ### Author Response · Authors · 2023-11-14
> **Reply to Reviewer C6xG**
>
> Dear Reviewer C6xG,
>
> We are very grateful to receive your positive feedback. Many thanks! We have carefully addressed your professional and insightful comments as below.
>
> **W1. The motivation of this work.**
>
> There is a consensus that studying complex dependencies in graph data is a major challenge for graph generation tasks[1, 2]. The most direct idea (solution) is to shield complex semantic association in graph, thus we naturally obtain the first research motivation, i.e., using subgraphs as generated elements (units). Specifically, the reason for using subgraphs as latent representation elements is that we have the following insights:
>
> - Given that whole graphs are with substantial and complex dependencies across numerous nodes, subgraph-level modeling can simplify most connections between nodes, and alleviate the challenges posed by complex dependencies in graphs.
>
> - Subgraphs can determine the property of the whole graph, such as functional groups can indicate the property of molecules. Taking subgraph as the minimum unit in graph diffusion process can enable an interpretable generative process for various science domains.
>
> Subsequently, the great success of latent (stable) diffusion model inspired us to propose the subgraph latent diffusion model. More importantly, subgraph-based diffusion generation pattern remains unexplored so far. Along this line of thinking, we conducted research on Subgraph Diffusion by proposing SubDiff.
>
>
> **W2. Subgraph-level generation paradigm and unified model.**
>
> As we discussed in Sec. 1, subgraph-level design not only provides a more interpretable generation process, but also empowers a great potential to integrate condition into unconditional models. Specifically, we argue that there is no stable causal (interpretable) relationship from nodes to graph properties. But for subgraphs, it has been proven that key subgraphs can potentially determine graph properties [3, 4]. Constraining strong prior hypothesis cannot facilitate the model to kill two birds with one stone, even may lead to inferior performance. Therefore, we take subgraph as a bridge to unify conditional-unconditional generation.
>
> In addition, we design node-level diffusion model NodeDiff (Sec. 6.3) to empirically answer whether node-level diffusion can also achieve a unified model with superior performance. We observe a  significant drop over performances, where it  even becomes weaker than the non-uniform node-level diffusion model (GEOLDM [1]).
>
>
> **W3. Missed related works.**
>
> As one of early works on graph latent diffusion models, NVDiff generates novel and realistic graphs by taking the VGAE structure and uses SGM as its prior for latent node vectors. More important, this work proven that the latent graph diffusion model has a proper lower-bound of the graph likelihood. NVDiff will be discussed in our manuscript.
>
> Thanks again for your constructive comments, and we will continue to improve our manuscript. Looking forward to your reply!
>
> **References:**
>
> [1] Geometric Latent Diffusion Models for 3D Molecule Generation, ICML 2023.
>
> [2] Generative diffusion models on graphs: Methods and applications, 2023.
>
> [3] Interpretable and Generalizable Graph Learning via Stochastic Attention Mechanism, ICML 2022.
>
> [4] On Explainability of Graph Neural Networks via Subgraph Explorations, ICML 2021.

---

> > ### Author Response · Authors · 2023-11-20
> > **Reply to Reviewer C6xG (2)**
> >
> > Dear Reviwer C6xG
> >
> > Thank you once again for dedicating your valuable time to reviewing our work. We have already submitted our response six days ago and sincerely hope it can address your concerns. We also note that that there are only 48 hours remaining for the discussion phase. Thus, we would like to know whether our response adequately addresses your questions, and if there are any additional inquiries you may have.
> >
> > We sincerely look forward to your response.
> >
> > Thanks!

---

### Official Review · Reviewer_Dfq7 · 2023-10-31

**Soundness:** 2 fair
**Presentation:** 2 fair
**Contribution:** 2 fair
**Rating:** 3
**Confidence:** 4

**Summary:**

This paper proposes a subgraph latent diffusion model for 3D molecular generation. Its main contributions are: 1. overcoming the dependency between nodes through subgraphs; and 2. proposing a unified model for both unconditional and conditional generation.

**Strengths:**

1.	A novel subgraph latent diffusion model is proposed in this paper.
2.	A unified framework is proposed for both unconditional and conditional generation.

**Weaknesses:**

1.	The authors first propose that the discrete geometric property of graphs makes it difficult to capture complex node-level dependencies for diffusion models. They claim that this problem can be solved by using subgraphs, which they present as the main contribution of this paper. I disagree with this viewpoint. Firstly, the existence of complex node-level dependencies has nothing to do with whether the data is discrete or continuous. Whether it is discrete atomic features or continuous positional features, complex node-level dependencies still exist. Secondly, while abstracting multiple nodes into subgraphs eliminates node dependencies, there can still be dependencies between subgraphs. However, the paper does not propose a solution for this subgraph dependency issue.
2.	The authors' second contribution is the proposal that subgraph latent embedding with explicit supervision can bridge the gap between unconditional and conditional generation. However, the explicit supervision used in the paper is graph-level label, and I do not get the contribution of subgraph latent embedding. In other words, the proposed solution in the paper, such as pooling subgraph latent embedding as in Eq. 7, could be replaced by pooling node latent embedding to obtain graph latent embedding. I doubt the necessity of using subgraph latent embedding to bridge the gap between unconditional and conditional generation.
3.	The description of the methods proposed in the paper is not clear enough. Two methods are proposed in the paper: subgraph-level equivariant architecture (SE-GNN) and head-alterable sampling strategy. Firstly, the paper lacks a clear explanation of how to implement L1 and L2 in SE-GNN. Secondly, in Section 4.2, the authors do not explain why it is called head-alterable, and it is not clear why this is considered a sampling strategy. From the beginning of page 6, this method changes the mean of the Gaussian distribution during the training phase. Additionally, the paper does not explain how to personalize the prior distribution for each property $y_i$, some equations to be presented.
4.	Some recent related works need to be compared, such as MDM [1].

[1] Huang, Lei, et al. "Mdm: Molecular diffusion model for 3d molecule generation." Proceedings of the AAAI Conference on Artificial Intelligence. Vol. 37. No. 4. 2023.

**Questions:**

Proposed in Weaknesses

---

> ### Author Response · Authors · 2023-11-14
> **Reply to Reviewer Dfq7**
>
> Dear Reviewer Dfq7,
>
> Thanks for your valuable comments to further improve our manuscript! Here we carefully address your concerns as follows.
>
> **W1. The explanation of the discrete geometry and subgraph-level dependencies.**
>
> Actually, the discreteness of graph refers to the characteristics of non-Euclidean geometry, while another opposite aspect is grid-based data which is associated with Euclidean geometry [1, 2, 3]. Thus, the discrete geometry here doesn’t indicate the discreteness or continuousness of numerical values. In our research, we specially investigate the diffusion process in the space of non-Euclidean geometry, and propose to take the advantage of subgraph learning to reduce the complexity of node-level dependencies along diffusion process.
>
> For the question of how to infer the subgraph-level dependencies, we dedicatedly design a SE-GNN in Sec 4.1 to aware the dependencies among subgraphs by maintaining the equivalence across them ((${L^2}$ component of SE-GNN ).
>
>
> **W2. Why the subgraph-level design makes sense?**
>
> The reason for using subgraphs as latent representation elements is that we have the following insights:
>
> - Given that whole graphs are with substantial and complex dependencies across numerous nodes, subgraph-level modeling can simplify most connections between nodes, and alleviate the challenges posed by complex dependencies in graphs.
>
> - Subgraphs can determine the property of the whole graph, such as functional groups can indicate the property of molecules. Taking subgraph as a minimum unit in graph diffusion process can enable an interpretable generative process for various science domains.
>
> Actually, since our generation task is to obtain graphs with desired properties, the explicit supervision in the work is exactly on graph-level labels. Overall, the contribution of subgraph latent embedding is to disentangle the complex dependencies in whole graphs and enable the interpretability for generative process.
>
> **W3. The explanation of SE-GNN and HAS.**
>
> **Insights and solution overview.** The graph diffusion process aims to construct distribution from training samples, and generates new graphs by sampling from constructed distribution. In our work, we establish two distributions as the separated sampling beginning heads to unify the conditional and unconditional generation task. We argue that different distributions can be controlled by different sampling beginnings, which results in various generated contents. In our work, we let the beginning distribution of conditional generation with supervision ($ H_C $) while unconditional generation sample from pure Gaussian distribution ($ H_U $).
> Concretely,
>
> - ${L^1}$ and ${L^2}$ are both basic backbones of GNNs to encode graph data, which is implemented by GIN in our work.
>
> - Based on above **Insights and solution overview**, we propose the ‘Head Alterable Sampling’ strategy, where ‘Head Alterable’ refers to the adjustable beginning (position), and it has been described in Sec. 5.2 and Fig. 1(c). We further show their generation (sampling) paths respectively:
>
> ${H_U}:z_{{G_S}}^{(T)} -  -  > \hat z_{{G_S}}^{(0)} -  -  > G$
>
> ${H_C}:z_{{G_S}}^{(0)} -  -  > z_{{G_S}}^{(T)} -  -  > \hat z_{{G_S}}^{(0)} -  -  > G$
>
>
> where $z_{{G_S}}^{(0)}$ ~ $N({\mu _i},I)$ and ${\mu _i}$ is pre-designed based on the characteristics of our focused properties as shown in Table 4.
>
> Thanks for your suggestion and we will incorporate these insights and description of our method into our manuscript.
>
> **W4. Missed related works.**
>
> Thank you for reminds. As a classic discrete graph diffusion model on molecule generation tasks, we will discuss Mdm [4] in detail in our manuscript.
>
> Your comments are constructive to our work. We will continuously improve our manuscript. Thanks again! Looking forward to your reply!
>
> **References:**
>
> [1] Digress: Discrete denoising diffusion for graph generation. ICLR 2023.
>
> [2] Structured denoising diffusion models in discrete state-spaces. NeurIPS 2021.
>
> [3] Geometric Latent Diffusion Models for 3D Molecule Generation. ICML 2023.
>
> [4] Mdm: Molecular diffusion model for 3d molecule generation. AAAI 2023.

---

> > ### Author Response · Authors · 2023-11-20
> > **Reply to Reviewer Dfq7 (2)**
> >
> > Dear Reviwer Dfq7
> >
> > Thank you once again for dedicating your valuable time to reviewing our work. We have already submitted our response six days ago and sincerely hope it can address your concerns. We also note that that there are only 48 hours remaining for the discussion phase. Thus, we would like to know whether our response adequately addresses your questions, and if there are any additional inquiries you may have.
> >
> > We sincerely look forward to your response.
> >
> > Thanks!

---

> > > ### Comment · Reviewer_Dfq7 · 2023-11-23
> > >
> > > Thank you for your response and explanation. However, I did not see any additions or modifications in the submitted paper. Moreover, I still maintain my opinion that if you emphasize subgraph as your motivation and innovation, you should provide more subgraph-level experiments to demonstrate its effectiveness. The Experiments section in the current version does not satisfy me. I will maintain my score.

---

> ### Author Response · Authors · 2023-11-23
> **Thanks for your follow-up**
>
> Dear Reviewer Dfq7,
>
> Thanks for your follow-up feedback. But we still would like to provide some necessary clarifications as follows.
>
> -	First, actually, to demonstrate the effectiveness of our motivation, our main experiments have focused on the subgraph-level graph representation learning, and have compared with non-subgraph-level approaches (e.g., GEOLDM). Besides, we have also degenerated SubDiff to a node-level latent diffusion model NodeDiff and make a necessary ablative comparison. Therefore, we believe that the experimental analysis of the advantages of subgraphs is sufficient.
>
> -	Secondly, we have responded/made clarifications to all questions and concerns you raised, but we have not got any interaction or feedback from any of the reviewers until now. Thus, we are unsure whether your concerns have been adequately addressed. Therefore, it doesn’t remain the opportunity to revise the paper. If the concerns have been addressed (or accepted), we will certainly make revisions in the next version based on suggestions of all reviewers accordingly.
>
> -	Last, in your previous reviews, you have not mentioned any additional experiments. In fact, based on the suggestions of Reviewer tXm3, we have already added some additional results about the setting of $ {\mu _i}$. Please refer to our revised submission.
>
> Finally, we are going to submit a revised version of this manuscript, and sincerely hope that the reviewers can provide an opportunity to consider our work in the next round of discussion period.
>
> Thanks again!